

# Use of spectral cloud emissivity to infer ice cloud boundaries: Methodology and assessment using CALIPSO cloud products

Hye-Sil Kim[1], Bryan A. Baum[2], and Yong-Sang Choi[1]

[1]Department of Climate and Energy Systems Engineering, Ewha Womans University, Seoul, Korea
[2]Science and Technology Corporation, Madison, Wisconsin, USA

*Correspondence to*: Yong-Sang Choi (ysc@ewha.ac.kr)

**Abstract.** Satellite-based operational cloud height retrievals generally assume a plane-parallel homogeneous cloud exists in each field of regard, or pixel, but this assumption ignores vertical inhomogeneity, which is of particular importance for optically thin, but geometrically thick, ice clouds. This study demonstrates that ice cloud emissivity uncertainties can be used to provide

a reasonable range of ice cloud layer boundaries, i.e., the minimum to maximum heights. Here ice cloud emissivity uncertainties are obtained for three IR channels centered at 11, 12, and 13.3 μm. The range of cloud emissivities is used to infer a range of ice cloud temperature/heights, rather than a single value per pixel as provided by operational cloud retrievals. Our methodology is tested using MODIS observations over the western North Pacific Ocean during August 2015. We estimate minimum/maximum heights for three cloud regimes, i.e., single-layer thin and thick ice clouds, and multi-layered clouds. Our

results are assessed through comparison with CALIOP Version 4 cloud products for a total of 11873 pixels. The cloud boundary heights for single-layer optically thin clouds show good agreement with those from CALIOP; bias for maximum (minimum) heights versus the cloud top (base) heights of CALIOP are 0.13 km (–1.01 km). For optically thick and multi-layered clouds, the biases of the estimated cloud heights from the cloud top/base become larger. Our method is applicable to measurements provided by most geostationary weather satellites including the GK-2A advanced multi-channel infrared imager.

The vertically resolved heights for ice clouds can contribute new information for studies involving weather prediction and cloud radiative effects.

## 1 Introduction

In this study, we develop an algorithm to infer cloud height for ice clouds using only infrared (IR) measurements for its applicability of global data regardless of solar illumination. Although the approach will be applied to geostationary satellites

in future work, the algorithm is developed for the Moderate Resolution Imaging Spectroradiometer (MODIS) sensor for two reasons: (1) our resulting cloud temperatures can be compared to those from the Cloud-Aerosol Lidar and Infrared Pathfinder Satellite Observation/Cloud-Aerosol Lidar with Orthogonal Polarization (CALIPSO/CALIOP) active lidar Version 4 products for verification and (2) further comparison can be made to the MODIS Collection 6 cloud products. The approach adopted in our study for the inference of ice cloud height has a basis in the work of Inoue (1985), who developed this approach using only



the split-window channels on the Advanced Very High Resolution Radiometer (AVHRR). The goal of the Inoue (1985) approach was to improve the inference of cloud temperatures for semi-transparent ice clouds. Heidinger and Pavolonis (2009) further improved this approach and generated a 25-year climatology of ice cloud properties from AVHRR analysis.

For satellite-based cloud height retrievals based on passive IR measurements, the radiative emission level is regarded as the cloud top. When the emissivity is 1, the cloud is emitting as a blackbody and the cloud top is at, or close to, the actual cloud's

upper boundary. As the emissivity decreases, the cloud top inferred from IR measurements will be lower than the actual cloud top level. This is demonstrated in Holz et al. (2006), who compare the cloud tops from aircraft Scanning High-Resolution Interferometer Sounder (S-HIS) measurements to those from co-incident measurements from the Cloud Physics Lidar (CPL). They find that the best match between the cloud tops based on the passive S-HIS measurements and the CPL occur when the integrated cloud optical thickness is approximately 1. This implies that the differences of cloud tops by IR measurements from

those by CALIOP are expected since the IR method reports the height where the integrated cloud optical thickness, beginning at cloud top and moving downwards into the cloud, is approximately 1 while CALIOP reports the actual cloud top to be where the first particles are encountered.

With regard to geometric differences of IR cloud tops from the actual cloud tops, optically thin but geometrically thick clouds show the largest bias, since the level of which the integrated optical thickness reaches 1 is much lower than the height

at which the first ice particles occur. In a review of different ten satellite retrieval methods for cloud top heights by IR measurements (Hamann et al., 2014), the heights inferred for optically thin clouds are generally below the cloud's mid-level height. When lower-level clouds are present below the cirrus in a vertical column, the inferred cloud height can be between the cloud layers, depending on the optical thickness of the uppermost layer.

To complement the use of IR window channels, the addition of a single IR absorption channel, such as one within the broad

15-µm $CO_2$ band, can improve the inference of cirrus cloud temperature (Heidinger et al., 2010). Their study shows how adding a single IR absorption channel at 13.3 µm to the IR 11- and 12-µm window channels decreases the solution space in an optimal estimation retrieval approach and leads to closer comparisons in cloud height/temperature with CALIPSO/CALIOP cloud products.

There are also retrieval approaches that use multiple IR absorption bands within the 15-µm $CO_2$ band (e.g., Menzel et al.

2008; Baum et al. 2012). Zhang and Menzel (2002) showed improvement of the retrieval of ice cloud height when they take into account spectral cloud emissivity that has some sensitivity to the cloud microphysics. As the goal of our work is to develop a reliable method for inferring ice cloud height from geostationary data, we are limiting this study to the use of the relevant IR channels, i.e., measurements at 11- 12-, and 13.3-µm.

Rather than infer a single ice cloud temperature in each pixel, we first infer a range of ice cloud temperatures (minimum to

maximum temperature per each ice cloud pixel) that correspond to uncertainties in the cloud spectral emissivity. We note that the spectral cloud emissivity, which can be obtained using measurements at 11, 12, and 13.3 µm, has some dependence on the ice cloud microphysics. The emissivities are used subsequently to estimate ranges of cloud height, which are found by converting the estimated cloud temperature ranges using a simple linear interpolation of the NWP model products. Cloud



boundary results are presented for three cloud categories, i.e., single-layer optically thin ice clouds, optically thick ice clouds, and multi-layered clouds, and these results are assessed with measurements from a month of collocated CALIOP Version 4 data. The focus area for the data analysis and resulting analyses is the western North Pacific Ocean for the month of August 2015.

The paper is organized as follows. Section 2 discusses the data and methodology and the generation of the relevant look-up tables (LUTs) for the radiances/brightness temperatures used in our analyses. Section 3 provides results for the western North Pacific Ocean during August 2015, and comparisons with CALIOP. Section 4 discusses the results and Section 5 concludes.

## 2 Data and Methodology

### 2.1 AQUA/Moderate Resolution Imaging Spectroradiometer (MODIS)

The MODIS is a 36-channel whisk-broom scanning radiometer on the NASA Earth Observing System Terra and Aqua platforms. The Aqua platform is in a daytime ascending orbit at 1330 LST. The MODIS sensor has four focal planes that cover the spectral range 0.42–14.24 μm. The longwave bands are calibrated with an onboard blackbody. The Aqua MODIS products used in this study include the Collection 6 1-km Level-1b radiance data (MYD021KM), geolocation data (MYD03), and the cloud properties at 1-km resolution (MYD06). The C6 MYD06 product provides cloud emissivity values in the IR window (8.5, 11, and 12 μm) and also cloud top height (CTH), all at 1-km spatial resolution; these parameters were not included in earlier collections (Menzel et al., 2008). The cloud emissivities at 11 and 12 μm are used in this study.

### 2.2 CALIPSO/CALIOP

The CALIPSO satellite platform carries several instruments, among which is a near-nadir-viewing lidar called CALIOP (Winker et al. 2007, 2009). Originally, CALIPSO flew in formation with NASA's Earth Observing System Aqua platform since 2006 and was part of the A-Train suite of sensors. At the time of this writing, it is no longer part of the A-Train but flies in formation with CloudSat in a lower orbit. CALIOP takes data at 532 and 1064 nm. The CALIOP 532-nm channel also measures the linear polarization state of the lidar returns. The depolarization ratio contains information about aerosol and cloud properties. This study uses CALIPSO Version 4 products that were released in November 2016. With the updated radiometric calibration at 532 and 1064 nm (Getzewich et al., 2018; Vaughan et al., 2019), cloud products such as cloud-aerosol discrimination and extinction coefficients show significant improvement relative to previous versions (Young et al., 2018; Liu et al., 2019).

### 2.3 Numerical weather model product

The Global Forecast System (GFS) model is produced by the National Centers for Environmental Prediction (NCEP) of the National Oceanic and Atmospheric Administration (NOAA) (Moorthi et al. 2001). GFS provides global Numerical Weather Prediction (NWP) model output at 0.5º resolution at 3-hour forecast intervals every 6 hours. We use two variables from the



NWP product, temperature profiles and geopotential heights, with cloud heights provided for 26 isobaric layers that are related

to cloud temperatures. The NWP fields are remapped to the resolution of satellite imagery and also interpolated to the time

corresponding to the satellite observation times.

**2.4 Cloud retrieval algorithm**

The basis for the retrieval algorithm is provided in Inoue (1985). Figure 1(a) shows the plane parallel homogeneous cloud

model with no scattering. The ice cloud layer at a given height has a corresponding ice cloud temperature ($T_c$) and an associated

cloud emissivity ($e_c$). The observed upwelling radiance ($I_{obs}$) at the cloud top is composed of two terms: the first depending on

the upwelling clear-sky radiance ($I_{clr}$) at the cloud base and the other depending on the radiance ($B(T_c)$) computed for a cloud

emitting as a blackbody:

$$I_{obs} = (1 - e_c)I_{clr} + e_c B(T_c), \tag{1}$$

where B($T_c$) is the Planck emission for a cloud computed at $T_c$ (Liou, 2002). All terms in Eq. (1) are wavelength dependent

except for the $T_c$. $I_{obs}$ is determined from the satellite measurements, and $I_{clr}$ can be found from clear-sky conditions in the

imagery or computed by a radiative transfer model given a set of atmospheric profiles of temperature, humidity, and trace

gases. However, $e_c$ and $T_c$ are unknown.

Eq. (1) can be rearranged to solve for the emissivity:

$$e_c = (I_{obs} - I_{clr})/(B(T_c) - I_{clr}). \tag{2}$$

One can relate two channels by taking a ratio of the radiances, similar to that of the $CO_2$ slicing method (e.g., Menzel et al.

2008) and assuming that the emissivity between two channels spaced closely in wavelength are the same. However, Zhang

and Menzel (2002) showed improvement of the retrieval of ice cloud pressure by accounting for differences in the spectral

cloud emissivity.

Inoue (1985) discusses the range of uncertainties in both $T_c$ and $e_c$ and further suggests that use of multiple IR channels can

reduce the uncertainties. To relate the effective emissivity between two channels, a parameterization is adopted that relates the

cirrus emissivity to the optical thickness. The $e_c$ is a function of the absorption coefficient ($\kappa$) and the cloud thickness ($z$),

$$e_c = 1 - exp^{-\kappa z/\mu} . \tag{3}$$

The term $\mu$ in Eq. (3) is a cosine of the viewing zenith angle; the quantity $\kappa z/\mu$ is called the optical thickness and is also

wavelength dependent. Given a value for $e_c$, the $T_c$ can be obtained by Eq. (2). The estimate of $e_c$ from an IR measurement

will have inherent uncertainties due to the diversity of ice particle size distributions (i.e., cloud microphysics), sensor

calibration, and in the cloud vertical inhomogeneity.



Another way to constrain these uncertainties is by using multiple IR channel measurements, specifically the spectral emissivity differences between two IR window channels ($\Delta e_c$). We can express the $\Delta e_c$ between two IR channels by:

$$\Delta e_c = exp^{-\frac{\kappa' z}{\mu}} - exp^{-\frac{\kappa z}{\mu}}. \tag{4}$$

In Eq. (4), $\kappa'$ is the absorption coefficient at 'another' IR window channel. That is, the $\Delta e_c$ is determined by $(\kappa - \kappa')/z$ which depends on the cloud particle size and cloud thickness (Kikuchi et al., 2006). Many studies have adopted this, or a similar, approach to apply the representative relations of spectral cloud emissivity relying on cloud types to retrieve the $T_c$ (e.g., Inoue, 1985; Parol et al., 1991; Giraud et al, 1997; Cooper et al, 2003; Heidinger and Pavolonis, 2009).

For the case of two IR channels, Inoue (1985) formulated the retrieval of the cirrus cloud temperature and effective

emissivity by setting up three equations with three unknowns (specifically referring to Inoue's equations 5, 6, and 7): Two equations are same as Eq. (2) at 11 and 12 μm in this paper, and the last equation is as follows.

$$e_c|_{12} = 1 - (1 - e_c|_{11})^{1.08}, \tag{5}$$

where $e_c|_{11}$ and $e_c|_{12}$ represent cloud emissivity for 11 and 12 μm. In Inoue (1985), the extinction coefficient between the 11- and 12-μm channels is set to a constant value of 1.08. The cloud temperature is determined by assuming a cloud emissivity at

one wavelength, calculating the emissivity at the other wavelength, and modifying the emissivities until a consistent cloud temperature  is found for both wavelengths. The initial assumed 11-μm cloud emissivity begins with a value of 0 and increases by a value of 0.01 until $T_c$ converges.

In this study, we design an approach as shown in Fig. 2 where more than two channels can be used. In this study, we examine the feasibility for estimating ice cloud temperature ranges, $\mathbf{T_c}$, given the uncertainties of the $e_c$ and $\Delta e_c$ (hereafter, $\mathbf{e_c}$ and $\Delta\mathbf{e_c}$)

such as $\mathbf{e_c} = [e_c^1, e_c^2, \cdots, e_c^n]$ and $\Delta\mathbf{e_c} = [\Delta e_c^1, \Delta e_c^2, \cdots, \Delta e_c^n]$ as shown in Fig. 1(b).

The differences between this study and Inoue (1985) are summarized as follows.

1. Constraints in the emissivity ranges ($\mathbf{e_c}$) in the different channels are provided in look-up tables (LUTs) discussed in the next section.

2. Emissivity differences are used, rather than a single value for the emissivity ratio between two channels.

3. Given the range of emissivity differences ($\Delta\mathbf{e_c}$ provided in LUTs), we obtain a range of $\mathbf{T_c}$ (and hence a range of cloud heights, $\mathbf{H_c}$) that can be compared to CALIPSO products.

The first step in the current method (Fig. 2) is to constrain 11-μm cloud emissivity for an ice cloud pixel and use $\mathbf{e_c}|_{11}$ values that are provided in LUTs (the light gray box in Fig. 2). The input parameters of LUTs are brightness temperature (BT) for 11 μm ($BT|_{11}$), BT differences (or BTD) between 11 and 13 μm ($BTD|_{11,13}$) and between 11 and 12 ($BTD|_{11,12}$).

The second step is to constrain cloud emissivity differences between 11 and 12 μm for an ice cloud pixel, $\Delta\mathbf{e_c}|_{11,12}$ that are also provided in LUTs (the dark gray box in Fig. 2) with identical input parameters as in the first step. Then the third step is to find $\mathbf{T_c}$ values that satisfy the three equations, i.e., a solution for Eq. (2) at both 11 μm and 12 μm and Eq. (4) using 11 and 12



µm with constraints in $\mathbf{e_c}|_{11}$ and $\Delta\mathbf{e_c}|_{11,12}$. The initial assumed 11-µm cloud emissivity begins with a value of $\min(\mathbf{e_c}|_{11})$ and increases by a value of 0.01 until $T_c$ converges. Notice that the $T_c$ value, an element of available ice cloud temperatures set, $\mathbf{T_c}$,

depends on $\Delta\mathbf{e_c}|_{11,12}$ in Eq. (4). That is, we obtain two $T_c$ values corresponding to min/max($\Delta\mathbf{e_c}|_{11,12}$). We define those two $T_c$ values as the minimum and maximum temperatures that an ice cloud pixel can have. Finally, we estimate cloud height ranges, $\mathbf{H_c}$, relating to min/max($\mathbf{T_c}$) by a dynamical lapse rate from GFS NWP temperature profiles provided for 26 isobar layers. In this study, any cloud height is not allowed to be higher than tropopause, which is provided in the GFS NWP model product.

## 2.5 Generation of look-up tables (LUTs)

For our method, relevant information for the western North Pacific Ocean is stored in look-up tables (LUTs). The LUTs include the min/max($\mathbf{e_c}$) and min/max($\Delta\mathbf{e_c}$) values for three indices: BTD$|_{11,13}$, BTD$|_{11,12}$, and BT$|_{11}$. The reason for selecting these three indices is that they are linked with cloud optical thickness, cloud effective radius, and cloud temperatures respectively. In fact, the vertical evolution in cloud microphysics has been investigated using passive satellite measurements, notably with using solar channels (e.g., Freud et al., 2008; Lensky and Rosenfeld, 2006; Martins et al., 2011). A primary benefit of using IR

measurements is that the ice cloud temperature and emissivity do not depend on solar illumination, so the cloud properties are consistent between day and night.

First, the BTD$|_{11,13}$ is sensitive to the presence of mid- to high-level clouds and the cloud height. While both the 12- and 13.3-µm measurements are both affected by $CO_2$ absorption, the 12 µm is at the wing of the broad 15-µm $CO_2$ band and has less $CO_2$ absorption than the 13.3-µm band. Additionally, the peak of weighting function for the 13.3-µm band is around 800

hPa so that the observed radiance at 13.3 µm represents middle and low atmosphere temperature. Thus, the BT at 13.3 µm is generally colder than that of the two other IR window channels. The BTD$|_{11,13}$ is larger for clear-sky pixels than for ice clouds, but BTD$|_{11,13}$ depends on degree of cloud opacity. The BTD$|_{11,13}$ has been applied by Mecikalski and Bedka (2006) to monitor changes in cloud thickness and height for signals of convective initiation.

Second, the BTD$|_{11,12}$ depends in part on the microphysics and cloud opacity, i.e., the number and distribution of the ice

particles; the imaginary part of the refractive index for ice varies in the IR region under study. The BTD$|_{11,12}$ has been used to identify cloud type (Inoue, 1985; Pavolonis and Heidinger, 2004; Pavolonis et al.,2005). Prata (1989) used the BTD$|_{11,12}$ to discern volcanic ash from non-volcanic ash. Recently, adding BTD from 8.6 and 11 µm, the BTD$|_{11,12}$ is also applied to infer cloud phase (Strabala et al.,1994; Baum et al., 2000, 2012).

Finally, BT$|_{11}$ values can provide cloud height information, at least for optically thick clouds including low-level clouds. For

optically thick clouds, the BT$|_{11}$ values approximate the actual cloud temperature, since at 11µm the primary absorber is water vapor and there is generally little absorption above high-level ice clouds. As noted earlier, the BT$|_{11}$ for optically thin clouds includes a contribution from upwelling radiances from the surface and lower atmosphere.

The LUTs are compiled for $\mathbf{e_c}$ and $\Delta\mathbf{e_c}$ by three input parameters, i.e., BTD$|_{11,13}$, BTD$|_{11,12}$, and BT$|_{11}$ from information in the C6 MODIS products. Data used in generating LUTs are summarized in Table 1. The first step is to collect all ice cloud

radiances centered at 11, 12, and 13.3 µm from MYD021KM over western North Pacific Ocean (0°N−30°N, 120°E−170°E)





during the recurring period of the August 2013 and 2014. Ice cloud pixels are identified by the MODIS IR cloud thermodynamic phase product in MYD06 that have a cloud top temperature ≤ 260K. The spatial and temporal domain is restricted to obtain a clear relationship between spectral cloud emissivity and three IR parameters for the case study analyses that will be presented later.

The second step is to categorize the ensemble of ice cloud pixels by three parameters, $BTD|_{11,13}$, $BTD|_{11,12}$, and $BT|_{11}$. The collected cloud pixels are separated into cloud types linked with cloud microphysical properties. We convert radiances centered at 11, 12, and 13.3 μm to BT by the inverse Planck's function and then calculate $BTD|_{11,13}$, $BTD|_{11,12}$, and $BT|_{11}$ for each pixel. Subsequently the ice cloud pixels are sorted into range bins defined for the three parameters as follows: $BT|_{11}$ values in a range from 190 K to 290 K in increment of 5K; $BTD|_{11,13}$ values in a range from –2 K to 30 K in increments of 2 K; and $BTD|_{11,12}$

values ranging from –1 K to 10 K in increments of 0.5 K (Table 2). For example, the first category is 190 K ≤ $BT|_{11}$ < 195 K, $-2 \leq BTD|_{11,13} < 0$, and $-1 \leq BTD|_{11,12} < -0.5$.

The final step is to generate LUTs based on an empirical relationship between cloud emissivity and $BTD|_{11,13}$, $BTD|_{11,12}$, and $BT|_{11}$, resulting in finding values for the min/max($\mathbf{e_c}$) and ($\Delta\mathbf{e_c}$) for each ice cloud pixel in each bin. The cloud emissivity values for each ice cloud pixel are provided in MYD06, for which Scientific Data Sets (SDS) names are 'cloud_emiss11_1km' and

'cloud_emiss12_1km', respectively. To exclude extreme values, the min/max($\mathbf{e_c}$) and ($\Delta\mathbf{e_c}$) are sorted into percentiles. The min/max($\mathbf{e_c}$) and ($\Delta\mathbf{e_c}$) are defined as the 2[nd] /98[th] percentiles for the ice cloud emissivity distribution for the various bins of the three parameters provides that there are at least 5,000 pixels available for a given bin. When there are between 500 and 5000 pixels, the 5[th] /95[th] percentiles are chosen as the min/max($\mathbf{e_c}$) and ($\Delta\mathbf{e_c}$). In the rare case when there are between only 200 and 500 pixels, the 10[th] /90[th] percentiles are used. Any case with fewer than 200 ice cloud pixel numbers is not included in the

LUTs.

Fig. 3 shows examples of LUT values for $\mathbf{e_c}$ belonging to the specific category for 230 K ≤ $BT|_{11}$ < 235 K (Fig. 3(a)) and 270 K ≤ $BT|_{11}$ < 275 K (Fig. 3(b)), which imply the presence of optically thick and thin ice clouds, respectively. The minimum (the left panel) and maximum (the right panel) values of the $\mathbf{e_c}$ are shown as colors in the space of $BTD|_{11,12}$ (x-axis) and the $BTD|_{11,13}$ (y-axis). In Fig. 3(a), the $\mathbf{e_c}$ values range from about 0.8 to 1.1. The $\mathbf{e_c}$ generally ranges from 0 to 1, but a non-physical

$\mathbf{e_c}$ value over 1 might occur in case of an over-shooting cloud (from strong convection that briefly enters the lower stratosphere) that has colder temperature than surrounding environment temperature (Negri, 1981; Adler et al., 1983). As for thin clouds, the $\mathbf{e_c}$ values of Fig. 3(b) range from around 0.3 to 0.8. In general, $\mathbf{e_c}$ values are low when cloudy pixels have large values of $BTD|_{11,12}$ and $BTD|_{11,13}$.

Fig. 4 shows examples of LUT values of $\Delta\mathbf{e_c}$ for optically thick (Fig. 4(a)) and thin (Fig. 4(b)) ice clouds as shown in Fig.

3. The $\Delta\mathbf{e_c}$ ranges from –0.12 to 0.04. The $\Delta\mathbf{e_c}$ shows a more complex relationship with $BTD|_{11,12}$ and $BTD|_{11,13}$ than $\mathbf{e_c}$ does. It is notable that similar patterns $\Delta\mathbf{e_c}$ are repeated on the optically thick (Fig. 4(a)) and thin ice cloud (Fig. 4(b)). One reason for this could be that $\Delta\mathbf{e_c}$ are more sensitive to particles sizes, whereas $\mathbf{e_c}$ values are more directly linked with cloud opacity (refer to Eq. (3) and Eq. (4)). The optically thin ice cloud cluster tends to be more sensitive to $BTD|_{11,12}$, showing larger variations of $\Delta\mathbf{e_c}$ than the thick ice cloud cluster.



## 3 Results

### 3.1 Study domain

Both case study and monthly analyses are performed over the Western North Pacific in August 2015. The analysis domain coincides with where MODIS data are used to populate the LUTs. In the Western North Pacific, the ice clouds can be generated from diverse meteorological conditions including frequent typhoons. Note that the typhoon 'Goni' formed on 13 August and dissipated on 30 August, 2015, affecting East Asia. Case studies involving Typhoon Goni scenes are provided in Section 3.3.

### 3.2 Clear-sky maps generated from MODIS

Much of the input and auxiliary data for the tests are available from the radiances and cloud products in the MODIS C6 (Table 2). The $I_{obs}$ and BT at 11, 12, and 13.3 μm (channels 31, 32, and 32) are taken from the C6 MYD021KM data. Two ancillary data products are also necessary. One is an estimate of $I_{clr}$ that is required in Eq. (2). The MODIS pixels identified as being clear-sky are used to generate a gridded clear-sky map, which is the second ancillary product required for our method. To simplify the generation of this map, the MODIS data are subsetted to 5 km resolution. Monthly composites of $I_{clr}$ at 0.1°×0.1° resolution were generated by choosing the maximum value among radiances for three months of August (2013–2015) in each 0.1°×0.1° grid box. Fig. 5a presents spatial distribution of $I_{clr}$ at 11 μm ($I_{clr}|_{11}$, Fig. 5(a)), from 8 to 11 W m$^{-2}$ μm$^{-1}$ sr$^{-1}$. Then Fig. 5b presents spatial distribution of differences of $I_{clr}|_{11}$ from $I_{clr}|_{12}$. Large differences are shown in the western region, near the Philippines (green-colored contours in Fig. 5). Ice cloud pixels are identified using the IR cloud phase product in the MYD06, which in turn uses the MODIS cloud mask product MYD35. For each ice cloud pixel selected, application of the methodology shown in Fig. 2 results in the min/max($T_c$).

### 3.3 Comparison of min/max Tc with CALIPSO for three granules

#### 3.3.1 A scene for single-layer thin cloud (19 August, 2015, at 0320 UTC)

Figure 6 is a scene analysis for single-layer thin ice clouds for a granule at 0320 UTC on 19 August, 2015. Fig. 6(a) is a MODIS false color image that captures Tropical Cyclone Goni. Note that the image is rotated 90 degrees left to simplify comparison with CALIPSO. The heavy pink line (Fig. 6(a)) is the south-to-north CALIPSO track at the closest time to the MODIS observation time. The CALIPSO made a near-eye overpass of the cyclone. The CALIOP track measures a cross section of the cyclone, from the eyewall to the outer bands. Fig. 6(b) is a cross section from CALIOP data (Table 3) at the time of the overpass, that shows the horizontal (x-axis) and vertical (y-axis at the left side) locations of all cloud layers. The CALIOP vertical feature mask (VFM) indicates the presence of randomly-oriented ice and horizontally-oriented ice (sky-blue), and water (orange) cloud phase in the scene. The y-axis at the right side are for two supplementary data shown as gray lines. The gray solid line is the CALIOP COT at 532 nm, for the opacity of ice clouds. The gray dashed line is the standard deviation of the MODIS $I_{obs}|_{11}$ (STD($I_{obs}|_{11}$)) on the collocated path with the CALIOP track, calculated over a 5 × 5 pixel array centered at each cloud pixel. The STD($I_{obs}|_{11}$) includes cloud feature information (Nair et al., 1998). For example, pixels at cloud edges or



fractional clouds have relatively large $STD(I_{obs}|_{11})$. The $STD(I_{obs}|_{11})$ values are used to filter overcast cloud pixels. The data in Fig. 6 are primarily of single-layer ice clouds with horizontal homogeneity as demonstrated by the low value of $STD(I_{obs}|_{11})$.

For comparison with CALIPSO, the min/max($\mathbf{T_c}$) are converted to max/min($\mathbf{H_c}$) and are shown from our method (blue/green circles) to the VFM in Fig. 6(b). Also provided is the MODIS CTH (black circles) for reference. For these comparisons, we converted temperature to height using a dynamical lapse rate from GFS NWP temperature profiles. When the cloud pixel temperature is colder than the tropopause temperature, it is changed to be that of the tropopause and is converted to the tropopause height provided by GFS NWP. The solid red line indicates where the CALIOP COT is about 0.5. This line is a reference for the position where the passive remote sensing retrievals will place the cloud (Holz et al. 2006; Wang et al., 2014), well known as the radiative emission level. The radiative emission level should be thought of more as a guideline since the matched COT values can be different depending on cloud types or algorithm methods. To determine this depth in the cloud layer, we integrated the extinction coefficient, CALIOP $Q_e$ (Table 3), from the top of the cloud downwards until the COT reached about 0.5. Hereafter, we call that layer as the effective emission layer, EEL.

Note that the max($\mathbf{H_c}$) (blue circles) is close to the top of clouds except in the region of cloud edges and the eye of Goni. Bias between the cloud top and the max($\mathbf{H_c}$) is 0.46 km, that is –4.5 K in the aspect of temperature. It is interesting that the max($\mathbf{H_c}$) corresponding to uncertainties of cloud emissivity tends to occur at or slightly above the cloud top as indicated by CALIPSO, higher than the EEL and MODIS CTH. The max($\mathbf{H_c}$) on the cloud edges and the eye of the Goni are scattered from the base of cloud mask and tropopause height. Those regions show relatively large $STD(I_{obs}|_{11})$ and small COT. The height of the min($\mathbf{H_c}$) (green circles) also follows to the base of clouds with bias $= -1.58$ km (10.61 K in the aspect of temperature), slightly lower than EEL and MODIS CTH.

### 3.3.2 A scene for single-layer thick cloud (19 August, 2015, at 1530 UTC)

The second case is the single-layer thick ice clouds (Fig. 7) at 1530 UTC on 19 August 2015. Here we show the $BT|_{11}$ image instead of RGB image (Fig. 7(a)) since this is a nighttime scene. Fig. 7(a) is also rotated 90 degrees left. For this overpass, CALIOP observed clouds farther away from the center of Goni, and inspection of the cross-section in Fig. 7(b) suggests that most of cloud pixels are optically thick with COT values higher than 5, about where the CALIOP signal attenuates, and have relatively low $STD(I_{obs}|_{11})$ as indicated by the gray solid/dashed line in Fig. 7(b). In the comparison with the CALIOP VFM, the max($\mathbf{H_c}$) tends to occur at or slightly below the cloud top as indicated by CALIPSO, still higher than the EEL and MODIS CTH. The bias for the max($\mathbf{H_c}$) from the top of clouds is 2.38 km (–13.22 K), which is larger than that of optically thin ice clouds. The min($\mathbf{H_c}$) is close to the MODIS CTH and the EEL, but bias for min($\mathbf{H_c}$) from the cloud base is larger than that of optically thin clouds, –2.69 km (19.40 K). The passive IR measurements have an upper COT limit as shown in earlier studies (Heidinger et al. 2009; 2010). The height boundaries from our method brackets both the CALIPSO measurements and the MODIS retrievals.



### 3.3.3 A scene for multi-layer cloud (8 August, 2015, at 0520 UTC)

The third case also involves a cross-section of Goni, but this scene is more complex in that there is evidence of both multi-layered and less homogeneous ice clouds on the southern boundary of the typhoon (Fig. 8a). Note that the STD ($I_{obs}|_{11}$) on the CALIPSO track show relatively large variances, compared to the previous two cases (Fig. 8(b)). The CALIOP COT is omitted given the high fluctuations in the values. In the region of 10ºN–20ºN, the max/min($H_c$) in this region are often outside the boundaries of the VFM. The max($H_c$) (blue circles) are scattered from near the second cloud layer to the top of the first cloud at the tropopause. Some pixels of the min($H_c$) (green circles) values are also outside the range of the VFM. There is more than one reason causing these increased scatters, including the fact that the uppermost cloud layer is optically thin (over half of all pixels have COT < 1.5) and there are indications of lower cloud layers. In the region of 20ºN–30ºN, clouds on the top layer are relatively thick (on average, COT = 3.5). In that case, heights of the max($H_c$) on the multi-layer pixels tend to be close to the EEL, which is much lower than the top of clouds. This is to be expected for the case of a geometrically thick but optically thin cloud. Note that the value of the min($H_c$) on the multi-layered cloud pixels sometimes reach almost to the second cloud layer, rather than near the first layer. Further thought needs to be given to these cases.

### 3.4 Comparison of max/min $H_c$ with CALIPSO for August 2015

In this section, the max/min($H_c$) is compared with the cloud top/base height (CTH/CBH) from CALIOP over the Western North Pacific during August 2015. The procedure to collocate CALIOP with MODIS is described in Nagle et al. (2009). First, we qualitatively examine the max/min($H_c$) with the cloud layer vertical cross-section from CALIOP/MODIS matchup files (Table 3) in Fig. 6-Fig. 8. Second, we quantitatively investigate the max/min ($H_c$) for all ice clouds against CALIOP CTH/CBH during the month. The extinction coefficients profiles, cloud phase and their quality flags, and the number of cloud layers are extracted from CALIOP and used in this analysis (Table 3).

The matchup data are filtered as follows: only ice cloud phase pixels are chosen that have the highest quality (CALIOP QC for cloud phase = 1), where CALIOP COT > 1.5 and STD($I_{obs}|_{11}$) from MODIS ≤ 1, which helps to remove cloud edges and fractional clouds. The relationship is investigated between the max/min($H_c$) and CALIOP CTH/CBH for three cloud regimes; (1) single-layer optically thin ice clouds, (2) optically thick ice clouds, and (3) multi-layer clouds where the uppermost layer is optically thin cirrus. The CALIOP/MODIS matchup clouds are separated into single-layer and multi-layered cloud groups using the number of layers found (NLF) from CALIOP (Table 3). The multi-layered cloud group includes two or more cloud layers, excluding single-layered clouds. Among single-layer cloud pixels, we define thin/thick cloud groups as CALIOP COT which is less/greater than 3.5, referring to the ISCCP cloud classification (Rossow et al., 1985; Rossow and Schifer,1999).

Fig. 9 shows the joint histogram of the max/min($H_c$) (y-axis of left/right panels) as a function of the CALIOP CTH/CBH (x-axis) for single-layer thin (Fig. 9(a)), thick (Fig. 9(b)), and multi-layer (Fig. 9(c)). Table 4 provides all statistical quantities for Fig. 9 as correlations (corr), differences of the mean value (bias), and root mean square differences (rmsd). Additionally, all statistical quantities in terms of temperature are in the unit of K and are given in the round brackets in Table 4. For single





layer clouds, the majority of max($H_c$) values are scattered about the one-to-one line. The statistical values are corr = 0.61, bias

= 0.13 km, rmsd = 0.91 for thin clouds. This implies that minimum value of cloud height ranges corresponding to $\mathbf{e_c}$ and $\Delta\mathbf{e_c}$

are close to the cloud top for single-layer clouds as determined from CALIOP.

However, the scatter is higher for optically thick clouds, with corr = 0.65, bias = 0.30 km, rmsd = 1.08 (Table 4). As for the

max($H_c$) for multi-layer clouds, the majority of scatter points are on the lower right side of the one-to-one line, with corr =

0.25, bias = 1.41 km, and rmsd = 2.64. The lowest correlation and the largest bias for multi-layer clouds, as expected given

the assumption of single layer clouds in our method.

The comparisons of the min($H_c$) (y-axis of right panels in Fig. 9) to the CALIOP CBH (x-axis) for all cloud categories show

relatively large correlations, at least over 0.48. Scatter points in three joint histograms for all cloud types are parallel to the

one-to-one line, but show negative biases implying higher heights than CALIOP CBT. As with the cases of the max($H_c$), bias

of the min($H_c$) increases from single-layer thin (–1.01 km), to thick (–1.71 km) and multi-layer (–4.64km) clouds.

**4 Discussion of Results**

The results in Figs. 6–9 show the comparisons of the ice cloud height ranges obtained based on the ice cloud emissivity

uncertainties with both MODIS C6 products and vertical cross sections of clouds from CALIOP. We investigated minimum

and maximum of the estimated ice cloud heights per each cloud pixel for three cloud regimes during August 2015; (1) single-

layer optically thin clouds, (2) optically thick ice clouds, and (3) multi-layer clouds.

Overall, the maximum values of the estimated ice cloud height ranges for single-layer thin/thick clouds show some skill in

comparison with the cloud tops from CALIOP: corr = 0.61/0.65, bias = 0.13/0.30 km. In particular, we note that the upper

height boundary for optically thin clouds derived from our method are very close to the geometric cloud tops. For multi-layer

clouds, the maximum heights are occasionally much lower than the uppermost cloud layer as observed by CALIOP, showing

the highest bias at 1.41 km. Higher biases are expected in our method given the assumption of single layer clouds in each pixel.

Additionally, the skill of our method decreases when the upper cloud layer is composed of optically thin and fractional clouds;

in some cases, the method cannot determine an emissivity range from the LUTs, which were generated for single-layer ice

clouds.

The minimum heights for single-layer thin clouds reach near the base of cloud, with corr = 0.83, bias = –1.01 km. However,

those for thick/multilayer, the biases became larger, at most –4.64 km. That is, the minimum heights for thick clouds became

much higher than the CALIOP-observed cloud bases. This indicates that the IR method has an optical thickness limitation and

is more useful for lower optical thicknesses, which has been noted previously (e.g., Heidinger et al. 2010). Even with large

biases of minimum heights, it is notable that correlation coefficients between minimum heights and the cloud base for all three

cloud regimes are sufficiently large, at least 0.48.

Fig. 10 shows the frequency of occurrence of biases, mean(CALIOP $H_c$) minus mean($H_c$), as a function of CALIOP COT

for the single-layer clouds during August 2015. Here mean(CALIOP $H_c$) is defined as simple average of upper and lower cloud

boundary such as 0.5·(CALIOP CTH + CALIOP CBH). The mean($H_c$) is also defined as 0.5·(max($H_c$)+min($H_c$)). In a





comparison of the MODIS cloud mask with CALIOP, Ackerman et al., (2008) noted that the cloud mask performs best at optical thicknesses above about 0.4. The lidar has a greater sensitivity to particles in a column than passive radiance measurements. Based on this consideration, we limited our results to those pixels where the COT ≥ 0.5 in x-axis of Fig. 10.

Fig.10 illustrates that our resulting single-layer ice clouds boundaries are comparable to CALIOP, showing slightly negative biases except the region of 'COT≤1.5'. The negative biases of mean($\mathbf{H_c}$) from CALIOP are mainly caused by two factors: (1) The min($\mathbf{H_c}$) values for all cloud regimes tend to be higher than geometric cloud base. (2) The max($\mathbf{H_c}$) values are sometimes slightly outside the actual cloud boundaries. Perhaps this is caused in part by the conversion of temperature to height using the NWP model product. Another source of error could be that the radiances have some amount of uncertainty that was not

considered in our methodology. The notable point is that heights for thin cirrus (1.5<COT≤3.5) show the lowest biases, which current algorithms by passive imageries have difficulties to produce accurately. Fig.10 also addresses the weakness of our method. In the region of COT≤1.5, biases of mean($\mathbf{H_c}$) from CALIOP are largest and positive. This region might be relevant to fractional clouds or cloud edges. We infer that relationship of cloud emissivity at 11 and 12 μm, the key controller in our method, might not be optimal in the fractional clouds or cloud edges, resulting in lower heights.

A limitation of this study is that the LUTs are generated for spectral emissivity using IR sensor observations and level-2 products that still have errors and uncertainties. It would be interesting to extend this preliminary research by generating LUTs for spectral emissivity using CALIOP, not IR sensors. If we can obtain more diverse ice cloud emissivity in vertical cloud thickness, it could result in improvements in the resulting cloud temperatures/height ranges. Also, the LUTs based on CALIOP data/products could be used to reduce errors in inferring cloud temperatures for multi-layer clouds.

## 5 Summary

The intent of our study is to demonstrate that ice cloud emissivity uncertainties, obtained from three IR channels generally available on various satellite-based sensors, can be used to estimate a reasonable range of ice cloud temperatures as verified through comparison with active measurements from CALIPSO. For satellite-based retrievals with heavy data volumes, the general assumption is that the cloud in any given pixel can be treated as plane parallel, which simplifies the retrieval algorithms.

However, for ice clouds and particularly optically thin ice clouds known as cirrus, the plane-parallel assumption breaks down because cirrus tends to have greater vertical inhomogeneity than lower-level water clouds. Cirrus are optically thin but are often geometrically thick. For these cases, the inference of a cloud-top temperature for a given measurement may not be optimal. In our approach, a range of spectral ice cloud emissivity is calculated from which is, in turn, used to infer a range of cloud temperatures. These temperatures are converted to heights and subsequently compared to active lidar measurements

provided by CALIPSO/CALIOP products.

This study provides a methodology to infer a range of spectral cloud emissivity for each cloud pixel. The range in emissivity represents uncertainty in the cloud microphysics to some degree. In our approach, we generate two LUTs for cloud emissivity at 11 μm and cloud emissivity differences between 11 and 12 μm using the brightness temperatures at 11, 12, and 13.3 μm. The 11-μm channel is a window channel where the primary absorption is caused by water vapor. The 12-μm channel is



impacted by both $H_2O$ or and $CO_2$, while the 13.3-μm channel has more absorption by $CO_2$ than by water vapor. The benefit of a method that relies of IR channels is that it does not depend on solar illumination, so the cloud heights can be obtained consistently between day and night.

We estimate a range of ice cloud temperature corresponding to the ice cloud uncertainty generated by three IR channels centered at 11, 12, and 13.3 μm by MODIS C6. The focus area is northwestern Pacific Ocean during August 2015. We verified

the estimated ranges of ice cloud temperature for three cloud categories, i.e., single-layer thin and thick ice clouds, and multi-layered clouds, against the vertical feature mask for CALIOP. We show that the minimum/maximum values for the estimated range of ice cloud heights agree with CALIPSO measurements fairly well for single-layer optically thin clouds. However, for optically thick and multi-layered clouds, the biases of the minimum/maximum values for those ranges from the cloud top/base became larger.

This approach can be applied to the new geostationary satellites, such as Himawari-8 (launched in 2015), GOES-16/17 (launched in 2016 and 2017), and GK-2A (launched in 2018). The new features of ice cloud temperatures from base to top by geostationary IR observation could contribute to improved accuracy of weather prediction and cloud radiative effects.

In future work, we intend to improve upon this methodology by developing lookup tables for spectral cloud emissivity uncertainty with CALIOP. Above all, it is required to study for global area for applying this method to the new geostationary

satellites. Also, further study is required to add more infrared channels to resolve more accurate spectral cloud emissivity uncertainties.

*Author contributions.* HSK built, tested, and validated the algorithm, and wrote the manuscript. BB contributed to complete the algorithm and to review/edit the manuscript carefully. YSC provided initial idea for the algorithm and guidance of this study. All authors were actively involved in interpreting results and discussions on the manuscript.

*Competing interests.* The authors declare no conflict of interest.

*Acknowledgments.* This work was supported by the "Development of Cloud/Precipitation Algorithms" project, funded by ETRI, which is a subproject of the "Development of Geostationary Meteorological Satellite Ground Segment (NMSC-2019-01)" program funded by the National Meteorological Satellite Center (NMSC) of the Korea Meteorological Administration (KMA).

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






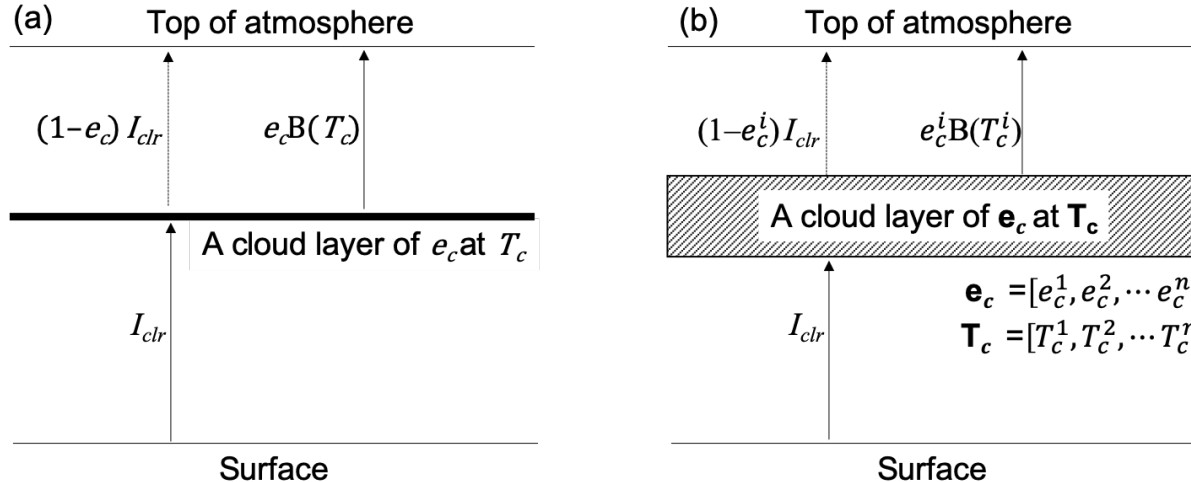

**Figure 1: Conceptual cloud model shown for (a) plane parallel homogeneous layer with cloud emissivity at 11 μm ($e_c$) and cloud emissivity differences between 11 and 12 μm ($\Delta e_c$) at the cloud temperature ($T_c$) and (b) plane parallel inhomogeneous layer with uncertainties in $e_c$ and $\Delta e_c$ such as $e_c = [e_c^1, e_c^2, \cdots, e_c^n]$ and $\Delta e_c = [\Delta e_c^1, \Delta e_c^2, \cdots, \Delta e_c^n]$ in a certain cloud temperature ranges, $T_c = [T_c^1, T_c^2, \cdots, T_c^n]$.**







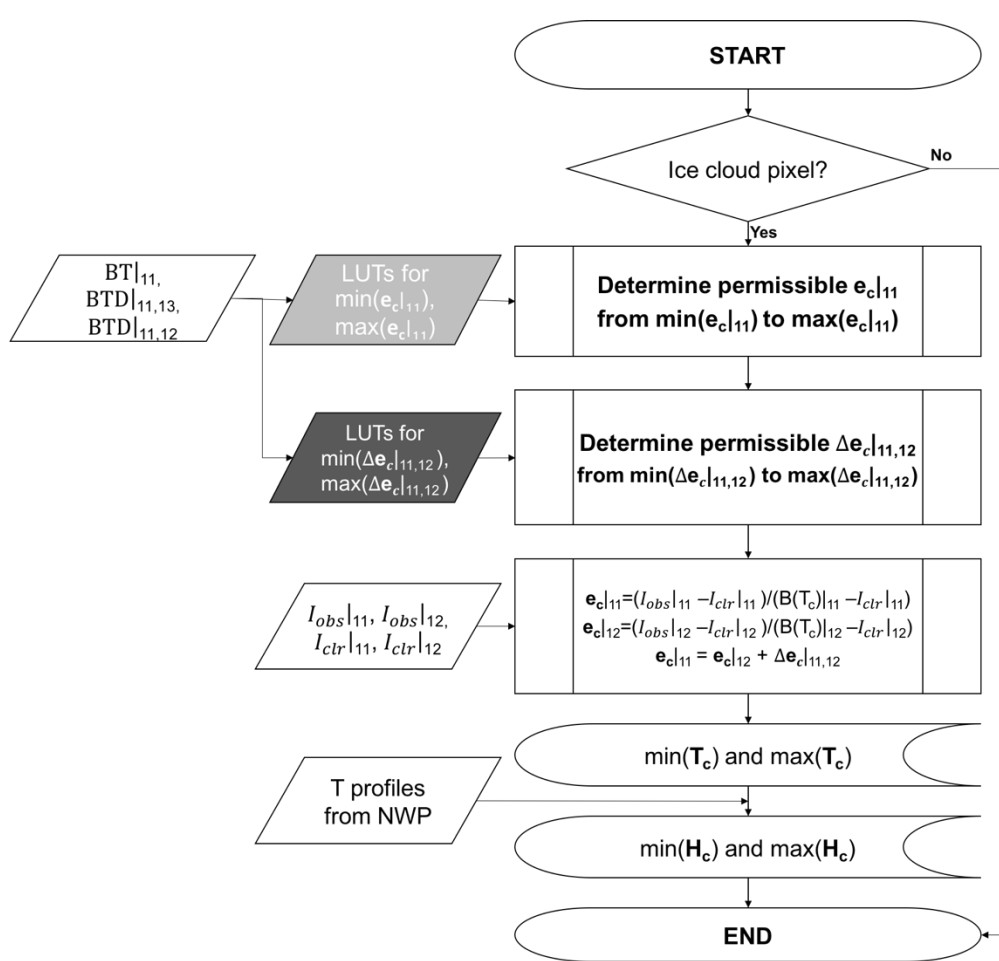

**Figure 2: The logo of Copernicus Publications. A flowchart for estimation of $T_c$ and $H_c$ corresponding to $e_c$ (from a light gray box that will be shown in Fig. 3) and $\Delta e_c$ (from a dark gray box that will be shown in Fig. 4) which represent cloud microphysics uncertainty in a certain cloud thickness. We denoted functions for minimum/maximum values of a matrix, A, as min/max(A).**








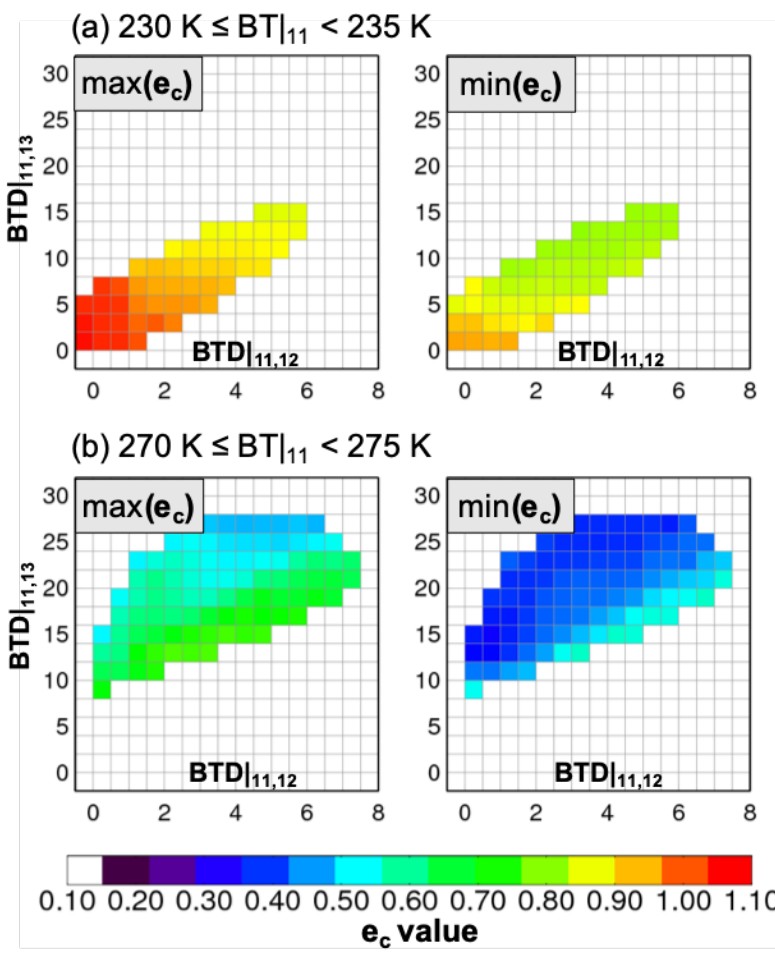


**Figure 3: Look-up table values for min/max($e_c$) (left/right panel in colors) by BTD$|_{11,12}$ (x-axis) and BTD$|_{11,13}$ (y-axis) for (a) 230 K ≤ BT$|_{11}$ < 235 K and (b) 270 K ≤ BT$|_{11}$ < 275K. For this look-up table, ice cloud pixels with temperatures ≤ 260 K were collected from MODIS C6 over the western North Pacific Ocean during two months of Augusts (2013– 2014). Table 1 summarizes data used in the look-up table. Also, Table 2 is for dimensions of the look-up table.**








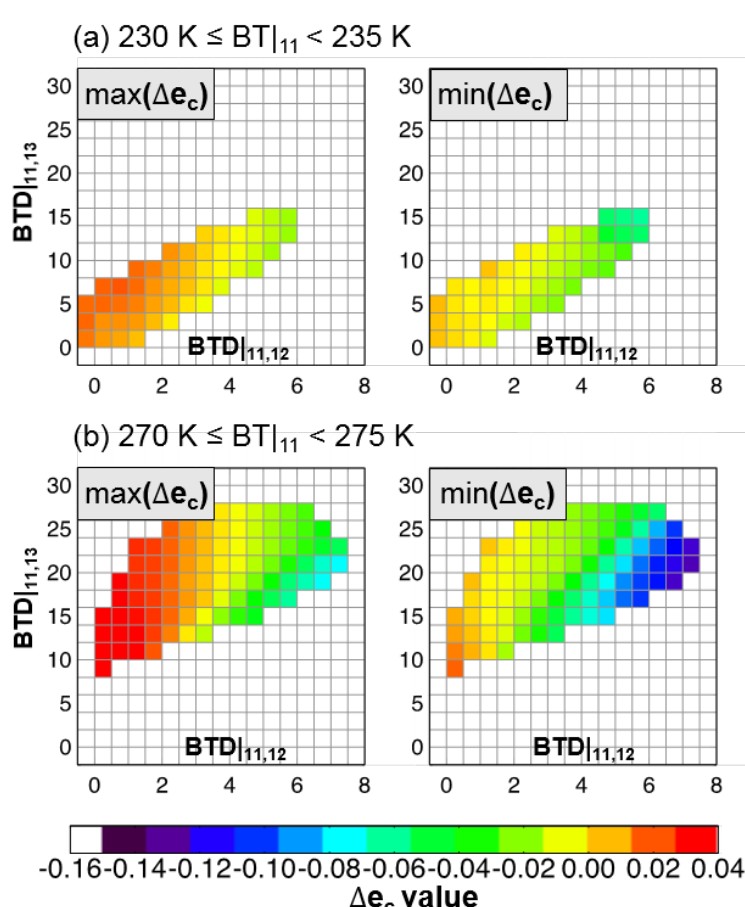


**Figure 4: Look-up tables for min/max($\Delta e_c$) (left/right panel in colors) by BTD|$_{11,12}$ (x-axis) and BTD|$_{11,13}$ (y-axis) for (a) 230 K ≤ BT|$_{11}$ < 235 K and (b) 270 K ≤ BT|$_{11}$ < 275K. Identical data as in Fig. 3 are used to generate these look-up tables, except cloud emissivity differences between 11 and 12 μm come from MODIS C6 (referring to Table 1 and Table 2).**







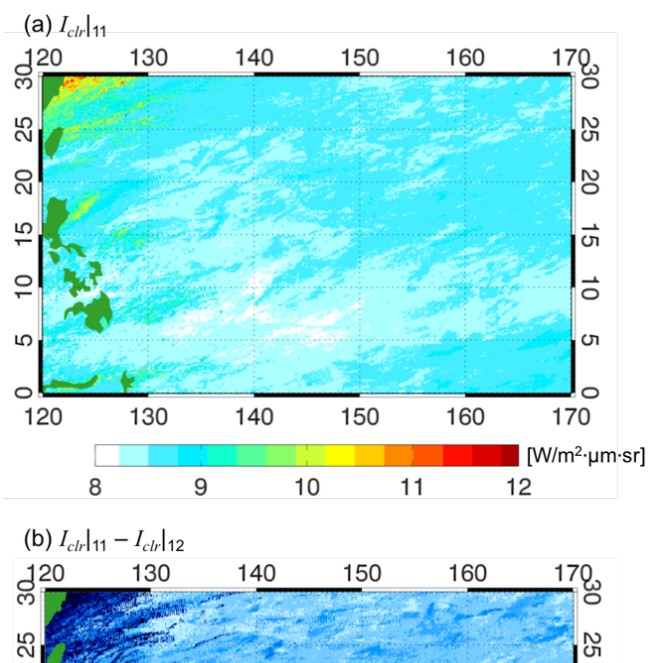


**Figure 5: The estimated clear sky radiance map at 0.1°×0.1° resolution for (a) 11 μm ($I_{clr}|_{11}$) in the unit of W m$^{-2}$ μm$^{-1}$ sr$^{-1}$. and (b) $I_{clr}|_{11} - I_{clr}|_{12}$. $I_{clr}|_{11}$ and $I_{clr}|_{12}$ are the maximum values among MODIS C6 radiances for three months of August (2013–2015) in each 0.1°×0.1° grid box. Green-shaded contours over the map show land, which is generally from the Philippines.**




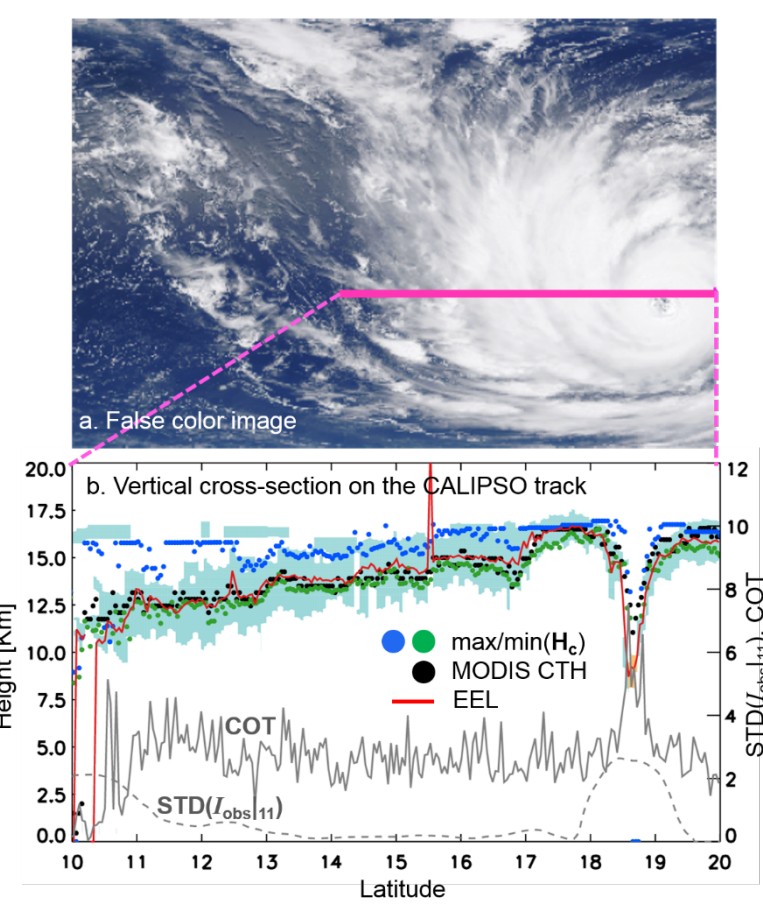


**Figure 6: (a) MODIS false color image (rotated 90 degrees left) at 0320 UTC 19 August 2015. This scene captures part of Typhoon Goni. The heavy pink line on the image shows CALIPSO track at the closest to MODIS observation time. (b) Vertical cross-section of the CALIPSO track designated by the heavy pink line in Fig. 6(a). The vertical feature mask is shown as sky-blue and orange contours (randomly and horizontally oriented ice, and water). The red solid line shows where the layer COT (integrated $Q_e$ at 532 nm from CALIOP) reaches a value of 0.5. The green/blue and black circles are the min/max($H_c$) and MODIS CTH, respectively. The gray solid (dashed) line on right side y-axis is the column COT from CALIOP (standard deviation of 11-μm radiances from MODIS.**





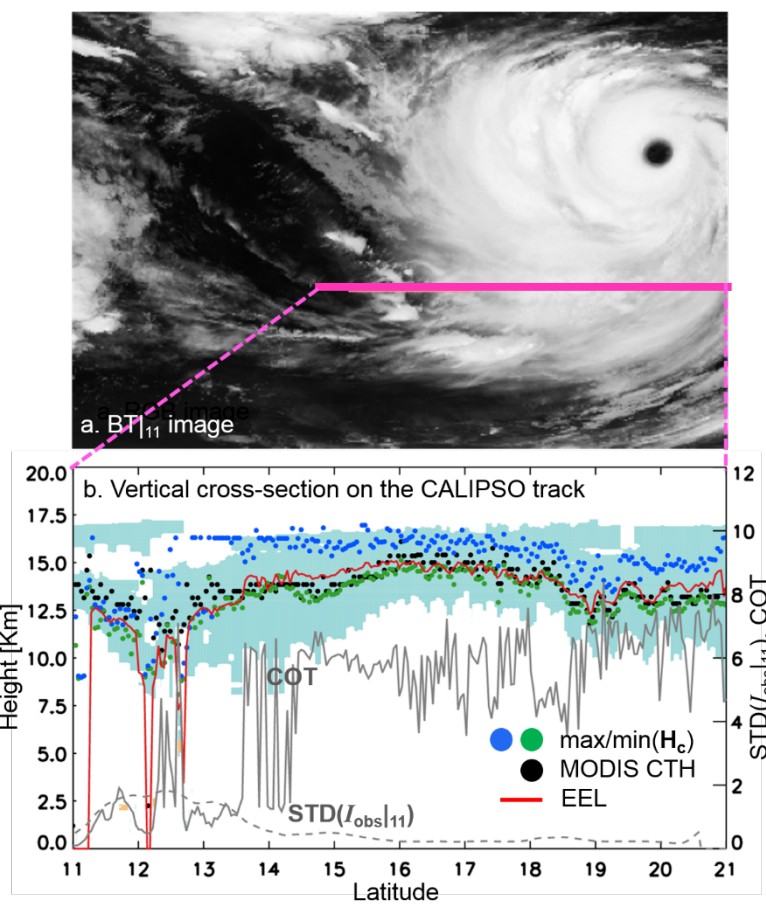

**Figure 7: (a) BT|$_{11}$ image from MODIS (MYD021 C6) at 1530 UTC on 19 August 2015. This scene captures part of Typhoon Goni.**
**The heavy pink line on the BT|$_{11}$ image shows CALIPSO track at the closest to MODIS observation time. (b) Vertical cross-section**
**of the CALIPSO track designated by the heavy pink line in Fig. 7(a). The vertical feature mask is shown as sky-blue and orange**
**contours (randomly and horizontally oriented ice, and water). The red solid line shows where the layer COT (integrated $Q_e$ at 532**
**nm from CALIOP) reaches a value of 0.5. The green/blue and black circles are the min/max($H_c$) and MODIS CTH, respectively.**
**The gray solid (dashed) line on right side y-axis is the column COT from CALIOP (standard deviation of 11-µm radiances from**
**MODIS.**



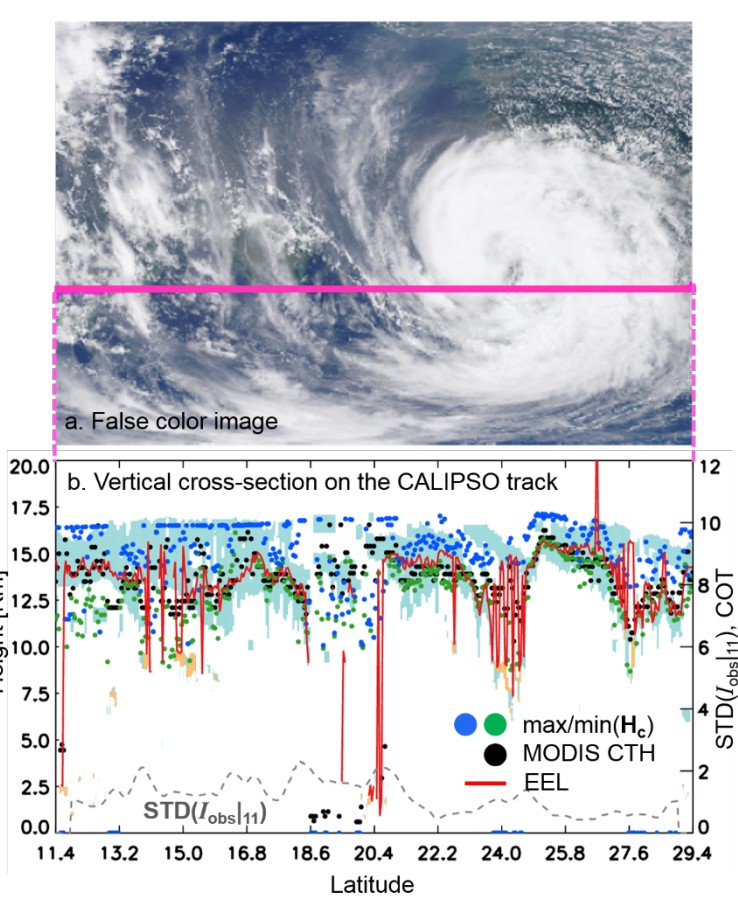


**Figure 8: (a) MODIS false color image (rotated 90 degrees left) at 0520 UTC on 8 August 2015. This scene captures part of Typhoon Goni. The heavy pink line on the image shows CALIPSO track at the closest to MODIS observation time. (b) Vertical cross-section of the CALIPSO track designated by the heavy pink line in Fig. 8(a). The vertical feature mask is shown as sky-blue and orange contours (randomly and horizontally oriented ice, and water). The red solid line shows where the layer COT (integrated $Q_e$ at 532 nm from CALIOP) reaches a value of 0.5. The green/blue and black circles are the min/max($H_c$) and MODIS CTH, respectively. The gray solid (dashed) line on right side y-axis is the column COT from CALIOP (standard deviation of 11-μm radiances from MODIS.**


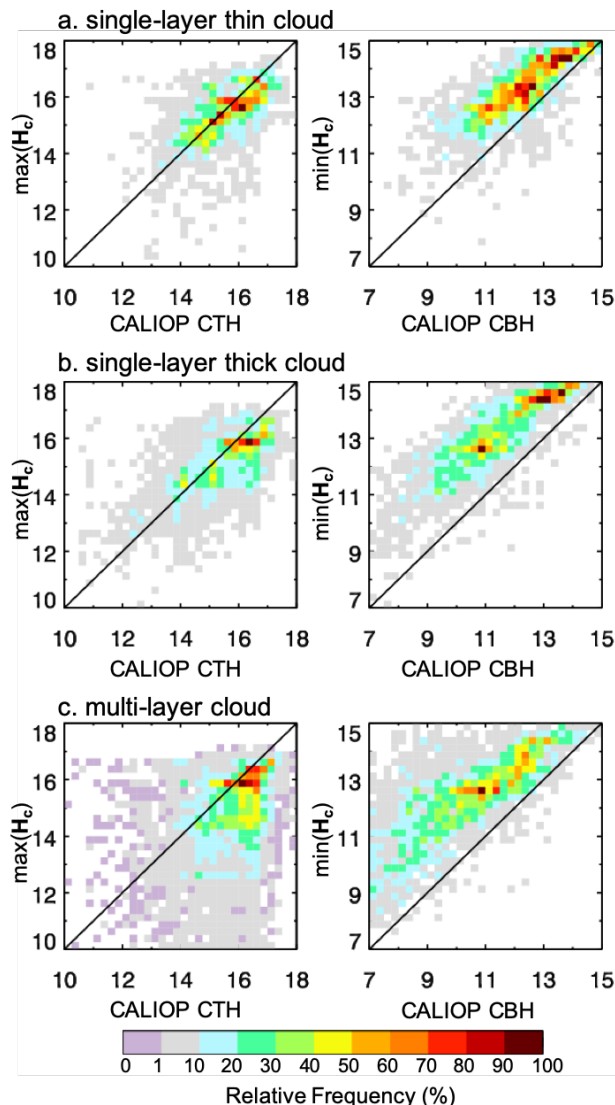

**Figure 9: Joint histograms of three cloud categories; (a) single-layer thin, (b) thick, and (c) multi-layer clouds during August 2015. The first column show CALIOP CTH (cloud top height, x-axis) versus max($H_c$) (y-axis), the second column shows CALIOP CBH (cloud base height, x-axis) versus min($H_c$) (y-axis).**



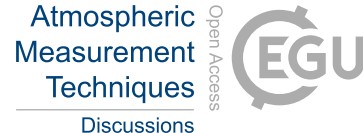



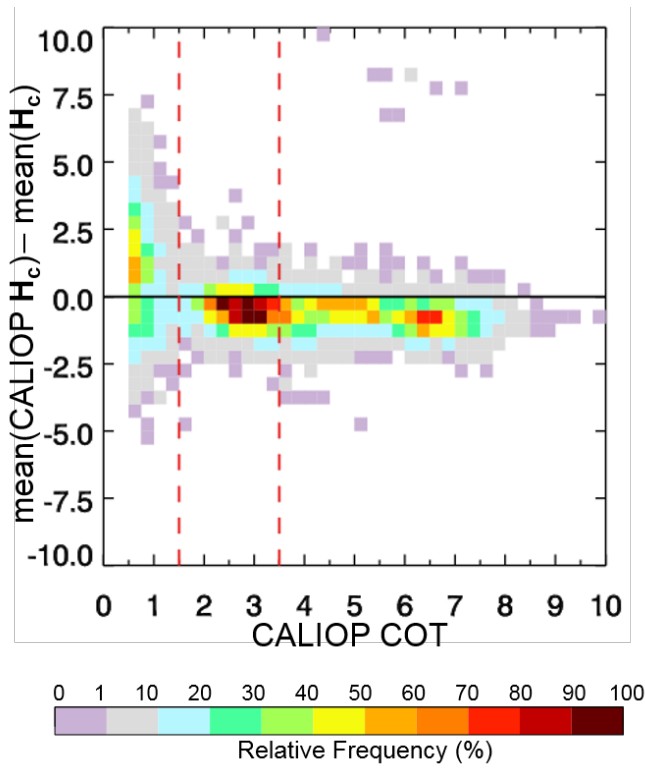


**Figure 10: A frequency of biases of mean($H_c$) from mean(CALIOP $H_c$) as a function of CALIOP COT during August 2015. The mean(CALIOP $H_c$) implies the average of upper and lower cloud boundary, simply defined as 0.5·(CALIOP CTH+CALIOP CBH). The mean($H_c$) is also the average of cloud heights by our method, defined as 0.5·(min($H_c$)+max($H_c$)). The red dotted lines are references for single-layer thin (1.5<COT≤3.5) and thick (COT>3.5) clouds in this study.**







| | Variables | MODIS C6 products | Period | Domain |
|---|---|---|---|---|
| Input data | $BT\|_{11}$ $BT\|_{12}$ $BT\|_{13}$ | Band 31, 32, and 33 in MYD021 | August 2013/2014 | Western North Pacific (0°N-30°N, 120°E-170°E) |
| Output data | min/max($e_c$) min/max($\Delta e_c$) | Cloud products in MYD06 | | |
| Auxiliary data | IR cloud phase | | | |

**Table 1: The detailed information used to generate empirical look-up tables (LUTs) of min/max($e_c$) and min/max($\Delta e_c$). MODIS bands 31, 32, and 33 have spectral wavelengths ranges of 10.78–11.28, 11.77–12.27, and 13.185–13.485 μm, respectively.**








| Input parameters | Value ranges | Increment |
|---|---|---|
| $BT|_{11}$ | 190 K – 290 K | 5 K |
| $BTD|_{11,13}$ | –2 K – 30 K | 2 K |
| $BTD|_{11,12}$ | –1K –10 K | 0.5 K |



**Table 2: Parameter ranges and discretization of parameters in the LUTs for $e_c$ (Fig. 3) and $\Delta e_c$ (Fig. 4)**







| | Variables | Data/products used in Sec. 3 | Period | Domain |
|---|---|---|---|---|
| Input data | $I_{obs}\|_{11}$ $I_{obs}\|_{12}$ $BT\|_{11}$ $BT\|_{12}$ $BT\|_{13}$ | Band 31, 32, and 33 in C6 MYD021 | August 2015 | Western North Pacific (0°N-30°N, 120°E-170°E) |
| Auxiliary data | $I_{clr}\|_{11}$ $I_{clr}\|_{12}$ | | August 2013/2014/ 2015 | |
| | IR cloud phase | Cloud products in C6 MYD06 | August 2015 | |
| | T/P profiles | GFS NWP products | August 2015 | |
| References for scene analysis | MODIS CTT/CTH | Cloud products in C6 MYD06 | August 2015 | |
| | VFM | CAL_L1D_L2_VFM-Standard-V4 | | |
| | CALIOP cloud phase | | | |
| | $Q_e$ | CAL_LID_L2_05kmCPro-Standard -V4 | | |
| | T/P profiles | | | |
| | COT | | | |
| References for statistical analysis | Number of layer found | CAL_LID_L2_05kmCLay-Standard-V4 | | |
| | CALIOP CTH/CBH | | | |
| | CALIOP CTT/CBT | | | |

**Table 3: Data used for the tests shown in Fig. 2. Input and auxiliary data are taken from the MODIS C6 cloud products and from CALIOP V4 cloud products. The abbreviations, CTT/CBT, CTH/CBH, COT, T/P, and VFM refer to cloud top/base temperature, cloud top/base height, cloud optical thickness, temperature/pressure, and vertical feature mask. The vertical profile of the extinction coefficient at 532 nm is denoted as the $Q_e$.**





| Category | Criteria | Count | CALIOP CTH vs. max($H_c$) (CALIOP CTT vs. min($T_c$)) | | | CALIOP CBH vs. min($H_c$) (CALIOP CBT vs. max($T_c$)) | | |
|---|---|---|---|---|---|---|---|---|
| | | | Corr | Bias | Rmsd | Corr | Bias | Rmsd |
| All ice | | 11873 | 0.31 (0.29) | 0.88 (− 6.15) | 2.07 (15.21) | 0.67 (0.70) | −3.17 (22.40) | 4.54 (30.20) |
| Single-layer thin cloud | NLF = 1 1.5 < COT ≤ 3.5 | 2237 | 0.61 (0.57) | 0.13 (−0.62) | 0.91 (6.12) | 0.83 (0.83) | −1.01 (8.02) | 1.31 (10.66) |
| Single-layer thick cloud | NLF = 1 COT > 3.5 | 3067 | 0.65 (0.66) | 0.30 (−1.53) | 1.08 (7.12) | 0.87 (0.87) | −1.71 (13.96) | 1.92 (15.53) |
| Multi-layer cloud | NLF > 1 | 6569 | 0.25 (0.23) | 1.41 (−10.18) | 2.64 (19.53) | 0.48 (0.48) | −4.64 (31.22) | 5.95 (38.69) |


**Table 4. Comparison of max($H_c$) (min($H_c$)) to the CALIOP CTH (CALIOP CBH) for all cloud pixels and three cloud regimes; single-layer optically thin ice clouds, thick ice clouds and multi-layer clouds for August, 2015. Pixel numbers (count), correlation coefficients (corr) and differences of the mean values (bias), root mean square differences (rmsd) are provided. Additionally, comparison of min/max($T_c$) to the CALIOP CTT/CBT are also shown as numbers of round brackets.**
