# Peer review of "Use of spectral cloud emissivity to infer ice cloud boundaries: Methodology and assessment using CALIPSO cloud products"

_Atmospheric Measurement Techniques, 2019_

## Referee Comment (RC1) · Carynelisa Haspel (Referee) · 16 Jun 2019

review of manuscript AMT-2019-148

In this manuscript, a relatively simple scheme for inferring the location of the upper and lower boundaries of clouds based mainly on infrared window channels is suggested and demonstrated. The idea is novel and as a suggested scheme for the remote sensing of clouds, it is highly relevant to Atmospheric Measurement Techniques. In general, the results of the new scheme are impressive and interesting. However, there are a number of clarifications that I suggest would be helpful to the reader, as follows.

(1) Given that at least ten different techniques already exist for retrieving cloud top heights with passive infrared radiation, it might be better to make the title more specific. Perhaps something like "A relatively simple scheme for inferring ice cloud boundaries using spectral cloud emissivity and its uncertainty..." would better emphasize the uniqueness of the current method as compared to the ones that already exist. See also my comment number 5.

(2) lines 7-8: "...generally assume a plane-parallel homogeneous cloud exists in each field of regard, or pixel, but this assumption ignores vertical homogeneity." – Strictly speaking, the plane-parallel assumption only ignores horizontal homogeneity within each pixel and *does* allow for vertical homogeneity. The more relevant argument here is that similar schemes additionally assume that the cloud is optically thin to the extent that there is only one value of cloud emissivity and only one value of cloud temperature per cloud in each pixel.

(3) line 14: "single-layer thin and thick ice clouds, and multi-layered clouds" – The distinction between these three categories is not precise. To be clear, I would write "single-layer thin ice clouds, single-layer [geometrically or optically?] thick ice clouds, and multi-layer clouds". This is true here and throughout the manuscript (e.g., lines 64-65).

(4) line 18: "become larger" – Cite how large here.

(5) Section 1 Introduction – In general, I think it would be helpful to the reader if the authors put the need for their scheme into better perspective. For example, it would be helpful to know from the outset under what circumstances non-window schemes, such as $CO_2$-slicing, can and cannot be used for the same purpose. Likewise, is the scheme suggested in the current study expected to provide an advantage over $CO_2$-slicing (a) because there an issue of availability of appropriate channels for implementing $CO_2$-slicing with wide enough spatial and temporal coverage, (b) because there is some issue of reliability with the $CO_2$-slicing technique, (c) because $CO_2$-slicing suffers from similar biases in that the cloud is assumed to be at a single altitude and to possess a single temperature, or (d) because the current scheme is just simpler? See also my comment number 14.

(6) lines 77-78: Briefly mention what scheme is used to retrieve cloud emissivity in the C6 MYD06 product.

(7) line 115: "a parameterization is adopted" – I believe that at these wavelengths, where scattering (by molecules) is negligible, $\exp(-\kappa z/\mu)$ is considered an accurate expression for the transmissivity based on the Beer-Lambert law and not a parameterization.

(8) line 118: "the quantity $\kappa z/\mu$ is called the optical thickness..." – I believe that $\kappa z$ is considered to be the optical thickness, rather than $\kappa z/\mu$.

(9) line 139: $\mathbf{T}_c$ is already in bold font, but it would be a good idea to emphasize in the text that this is a vector of possible values of cloud temperature rather than a single value.

(10) lines 151-152: In other words, one does not need to assume a ratio of cloud optical depths between the two channels, such as the ratio 1.08 in Equation 5, or is there some similar implicit assumption?

(11) line 155: "we obtain two $T_c$ values..." – Actually, a list of possible $T_c$ values, including the minimum and maximum possible values, is obtained, correct?

(12) lines 156-157: "... by a dynamical lapse rate..." – Does this mean that it is assumed that $T_c$ varies within the cloud layer, or does the lapse rate only apply to the atmospheric layers outside of the cloud layer?

(13) line 158: "any cloud height is not allowed..." – "no cloud height is allowed..."

(14) lines 163-166: "In fact, ... day and night." – These sentences belong in the Introduction. Refer also to my comment number 5.

(15) lines 97-98: "... based on an empirical relationship..." – Does this mean that the authors' Equation 4 is not used, or is the empirical relationship related to Equation 4?

(16) lines 264-266: "It is interesting that..." – I think that this should be emphasized better. It is not an interesting side note but an impressive demonstration of the concept suggested in this paper.

(17) line 266: "... and the eye of the Goni..." – It does not seem that there are data points right in the eye, only at the edges of the eye, correct? Also "scattered from" is not the best wording here and throughout the discussion.

(18) lines 267-269: Again, I think the success of the authors' method should be emphasized better here. These sentences explain the difference between the results of their method and the other data in the regions of the cloud edges and the eye of the hurricane. However, given that the left side of the image appears to contain multiple cloud layers that are likely moving, the fact that the authors' results and the CALIOP VFM data exhibit similar variation and similar values near the tropopause actually demonstrates a rather decent qualitative correspondence between the two.

(19) lines 278-279: Once again, I think the success of the authors' method should be emphasized better here. The bias for $\min(\mathbf{H}_c)$ from the cloud base is larger than that of optically thin clouds, but it is still better than the EEL, which is what would have been predicted.

(20) line 315: "... minimum value" – Was "maximum value" intended here?

(21) Figures 6-8: The orange contours are barely visible. Perhaps they are not necessary to include. Also, a closing parenthesis seems to be missing from the end of each of the three figure captions.

---

## Referee Comment (RC2) · Anonymous Referee #1 · 1 Jul 2019

General comments: This paper described a method to estimate the height of ice clouds from satellite measurements of three infrared channels. The paper is well written and includes useful information for researchers in the field of satellite remote sensing. However, it was difficult to understand how the vertical inhomogeneity of ice clouds was considered in this method. The reviewer concluded that additional explanation is necessary in the manuscript before AMT publication. Specific comments are addressed below.

Page 5 line 138-140 and Fig.1: What does $e\_c\hat{}i$ (i=1âŃŕn) in Fig.1b mean? Why $e\_c$ and $T\_c$ represent inhomogeneous layer? In the reviewer's understanding, $e\_c$ (and

T_c) describes a range of possible cloud emissivity (and temperature) that can simulate the measured channel radiances assuming a homogeneous cloud layer.

Page7 line 197-200: I suppose that the LUT for the empirical relationship between cloud emissivity and BT/BTD is a key of the proposed method. Does the author assume the dataset MYD021KM and MYD06 provide the relationship for vertically inhomogeneous cloud layer? I think that the author should express the basic idea of your approach for inhomogeneous cloud layer in the manuscript.

Page 8 line 233-237 and Fig.5: The reviewer cloud not understand what does the author intend to show in Fig.5a and 5b. What does the region of large differences of I_(clr|11)-I_(clr|12) in Fig.5b suggest?

Page 22 Fig.6b and Page 8 Fig.8b: What is the enhancement of EEL at latitude 15.6° of Fig. 6b? Similar enhancement is also appeared at latitude 25.7° in Fig. 8b.

---

## Referee Comment (RC3) · Anonymous Referee #3 · 9 Jul 2019

General comments This study uses spectral cloud emissivity to derive information regarding the minimum and maximum values of cloud top height (CTH). Authors primarily use MODIS data to derive the relationship between brightness temperature (BT) or brightness temperature difference (BTD) and emissivity values to infer information of cloud top temperature (CTT), and then convert CTT into CTH. They used CALIPSO data to validate their products. Though such type of study is essential to improve our understanding regarding CTH retrieval accuracy by MODIS and other satellite sensors, this study needs more improvement to full this gap as explained in detail in the specific comments below. The present version of the manuscript needs substantial revision. The presentation is not clear and discussion is relatively poor. The study method (Figure 2) is ambiguous. For example, what information do authors use from ice cloud pixels to determine the permissible ec ?, what is the meaning of permissible ec ?, do authors use emissivity data or uncertainty in emissivity? There are a number of such confusions to the reader. Further, It is not clear how this study can address the problem of cloud vertical inhomogeneity as stated in the first line of abstract. It should be either removed or discussions are necessary to show how this study can address such problem. The discussions presented in the second half are relatively poor. For example, what are authors' view for relatively large difference in min(Hc) and CALIOP base height in Figure 9?. The English also needs to be improved.

Specific comments 1. L63: Write the full form of NWP as it appears for the first time here. 2. Section 2 :It is better to separate data and methodology in different sections. 3. L95: Specify what method is used while remapping NWP fields to the resolution of satellite imaginary and interpolating to the time corresponding to satellite observation. 4. L140: Are ec and ấŰş ec are obtained are SDS data of 'cloud_emiss11_1km' and 'cloud_emission12_1km' as expressed in L200. Are they the emissivity or emissivity uncertainties? If 5. L140:L155: Make this section clear and easy to understand. For example, how do you constrain 11 micron cloud emissivity for an ice cloud pixel (L147), and how do you use this information with LUT values? 6. L197-L204: This paragraph is also confusing. The first line of this paragraph states that you derive an empirical relationship, however, the last section discusses about taking percentile values. Do you use empirical relationship or percentile values to define the minimum and maximum values of the emissivity? 7. Subsections 3.1 and 3.2 may be moved to data section. 8. L297: A brief description regarding the procedure of collocating CALIOP and MODIS is useful here. 9. What are authors' view for deviated CBH and min(Hc)? 10. Why not to write min_CTH or similar instead of min(Hc) ? Same for max (Hc) as well. 11. The discussion of section 4 may be strengthen by referring past studies and/or putting authors' own logic. 12. It is better to show the dependence of CTH or CBH difference between CALIOP and this study on CALIOP COT in Figure 10 instead of the mean value difference. What information do authors want to convey from the difference of

mean values? 13. Figure 1:Make the caption clear. Write about Iclr and B in the caption. 14. Figure 2: 'The logo of Copernicus Publications' should be removed from the caption. 15. If COT is not used here, why do you use COT for y-axis title ? 16. Table 1: What is IR cloud phase here ? 17. Table 2: Why 700 and 705 appear in this table?
* * *

---

## Author Comment (AC1) · 29 Jul 2019

We appreciate for your detailed review to strengthen our manuscript. I have a question about your comments, # 4. I think the last sentence is likely to disappear due to a systematic error. Could you please give the comment, #4 again? It will be helpful to revise the manuscript.

---

## Referee Comment (RC4) · Anonymous Referee #3 · 31 Jul 2019

**Comments on "Use of spectral cloud emissivity to infer ice cloud boundaries: Methodology and assessment using CALIPSO cloud products" by Kim et al.**

**General comments**

This study uses spectral cloud emissivity to derive information regarding the minimum and maximum values of cloud top height (CTH). Authors primarily use MODIS data to derive the relationship between brightness temperature (BT) or brightness temperature difference (BTD) and emissivity values to infer information of cloud top temperature (CTT), and then convert CTT into CTH. They used CALIPSO data to validate their products. Though such type of study is essential to improve our understanding regarding CTH retrieval accuracy by MODIS and other satellite sensors, this study needs more improvement to full this gap as explained in detail in the specific comments below. The present version of the manuscript needs substantial revision. The presentation is not clear and discussion is relatively poor. The study method (Figure 2) is ambiguous. For example, what information do authors use from ice cloud pixels to determine the permissible $e_c$ ?, what is the meaning of permissible $e_c$ ?, do authors use emissivity data or uncertainty in emissivity? There are a number of such confusions to the reader. Further, It is not clear how this study can address the problem of cloud vertical inhomogeneity as stated in the first line of abstract. It should be either removed or discussions are necessary to show how this study can address such problem. The discussions presented in the second half are relatively poor. For example, what are authors' view for relatively large difference in min(Hc) and CALIOP base height in Figure 9?. The English also needs to be improved.

**Specific comments**
1. L63: Write the full form of NWP as it appears for the first time here.
2. Section 2 :It is better to separate data and methodology in different sections.
3. L95: Specify what method is used while remapping NWP fields to the resolution of satellite imaginary and interpolating to the time corresponding to satellite observation.
4. L140: Are $e_c$ and $\triangle e_c$ are obtained are SDS data of 'cloud_emiss11_1km' and 'cloud_emission12_1km' as expressed in L200. Are they the emissivity or emissivity uncertainties? If
5. L140:L155: Make this section clear and easy to understand. For example, how do you constrain 11 micron cloud emissivity for an ice cloud pixel (L147), and how do you use this information with LUT values?
6. L197-L204: This paragraph is also confusing. The first line of this paragraph states that you derive an empirical relationship, however, the last section discusses about taking percentile values. Do you use empirical relationship or percentile values to define the minimum and maximum values of the emissivity?
7. Subsections 3.1 and 3.2 may be moved to data section.
8. L297: A brief description regarding the procedure of collocating CALIOP and MODIS is useful here.
9. What are authors' view for deviated CBH and min(**Hc**)?
10. Why not to write min_CTH or similar instead of min(**Hc**) ? Same for max (**Hc**) as well.
11. The discussion of section 4 may be strengthen by referring past studies and/or putting authors' own logic.

12. It is better to show the dependence of CTH or CBH difference between CALIOP and this study on CALIOP COT in Figure 10 instead of the mean value difference. What information do authors want to convey from the difference of mean values?

13. Figure 1:Make the caption clear. Write about $I_{clr}$ and B in the caption.

14. Figure 2: 'The logo of Copernicus Publications' should be removed from the caption.

15. If COT is not used here, why do you use COT for y-axis title ?

16. Table 1: What is IR cloud phase here ?

17. Table 2: Why 700 and 705 appear in this table?

---

## Author Comment (AC2) · 6 Aug 2019

review of manuscript AMT-2019-148

In this manuscript, a relatively simple scheme for inferring the location of the upper and lower boundaries of clouds based mainly on infrared window channels is suggested and demonstrated. The idea is novel and as a suggested scheme for the remote sensing of clouds, it is highly relevant to Atmospheric Measurement Techniques. In general, the results of the new scheme are impressive and interesting. However, there are a number of clarifications that I suggest would be helpful to the reader, as follows.

[Figure]

We greatly appreciate your detailed comments, which we used to revise and improve our paper as shown below.

(1) Given that at least ten different techniques already exist for retrieving cloud top heights with passive infrared radiation, it might be better to make the title more specific. Perhaps something like "A relatively simple scheme for inferring ice cloud boundaries using spectral cloud emissivity and its uncertainty..." would better emphasize the uniqueness of the current method as compared to the ones that already exist. See also my comment number 5.

Original title: Use of spectral cloud emissivity to infer ice cloud boundaries: Methodology and assessment using CALIPSO cloud products

Potential new title: A relatively simple scheme for inferring ice cloud boundaries using spectral cloud emissivity and its uncertainty: Methodology and assessment using CALIPSO cloud products

A second potential new title: Inference of cirrus layer boundaries using spectral cloud emissivity and its uncertainty: Methodology and assessment using CALIPSO cloud products

A third potential new title: Use of spectral cloud emissivities and their related uncertainties to infer ice cloud boundaries: Methodology and assessment using CALIPSO cloud products

In the revised version, we chose the third potential new title, 'Use of spectral cloud emissivities and their related uncertainties to infer ice cloud boundaries: Methodology and assessment using CALIPSO cloud products', since the spectral cloud emissivities and their uncertainties are unique factors to infer ice cloud boundaries in this paper.

(2) lines 7-8: "...generally assume a plane-parallel homogeneous cloud exists in each field of regard, or pixel, but this assumption ignores vertical homogeneity." – Strictly speaking, the plane-parallel assumption only ignores horizontal homogeneity within

each pixel and does allow for vertical homogeneity. The more relevant argument here is that similar schemes additionally assume that the cloud is optically thin to the extent that there is only one value of cloud emissivity and only one value of cloud temperature per cloud in each pixel.

Response: We agree that the relevant argument here is that even when optically thin ice clouds are present, operational retrievals provide only one value of cloud emissivity and one each of height/temperature/pressure per pixel. The text will be changed to reflect that we are assuming horizontal homogeneity with a given pixel but allowing for vertical inhomogeneity (that is, finding cloud boundaries).

[lines 8–12] "Satellite imager-based operational cloud property retrievals generally assume that a cloudy pixel can be treated as being plane-parallel with horizontally homogeneous properties. This assumption can lead to high uncertainties in cloud heights, particularly for the case of optically thin, but geometrically thick, ice clouds. This study demonstrates that ice cloud emissivity uncertainties can be used to provide a reasonable range of ice cloud layer boundaries, i.e., the minimum to maximum heights."

(3) line 14: "single-layer thin and thick ice clouds, and multi-layered clouds" – The distinction between these three categories is not precise. To be clear, I would write "single-layer thin ice clouds, single-layer [geometrically or optically?] thick ice clouds, and multi-layer clouds". This is true here and throughout the manuscript (e.g., lines 64-65).

Response: We clarified the three categories, "single-layer optically thin ice clouds, single-layer optically thick ice clouds, and multi-layer clouds" as follows throughout the manuscript.

[lines 15–16] "We estimate minimum/maximum heights for three cloud regimes, i.e., single-layered optically thin ice clouds, and single-layered optically thick ice clouds, and multi-layered clouds."

[Figure]

[lines 79–81] "Cloud boundary results are presented for three cloud categories, i.e., single-layered optically thin ice clouds, single-layer optically thick ice clouds, and multi-layered clouds, and these results are assessed with measurements from a month of collocated CALIOP Version 4 data."

the subtitle of section 4.1 [line 288] 4.1.1 A scene for single-layered optically thin ice cloud (19 August, 2015, at 0320 UTC) [line 323] 4.1.2 A scene for single-layered optically thick ice cloud (19 August, 2015, at 1530 UTC)

[line 289] "Figure 6 is a scene analysis for single-layered optically thin ice clouds for a granule at 0320 UTC on 19 August, 2015."

[line 324] "The second case is the single-layered optically thick ice clouds (Fig. 7) at 1530 UTC on 19 August 2015."

[lines 368–370] "Fig. 9 shows the joint histogram of the max/min(Hc) (y-axis of left/right panels) as a function of the CALIOP CTH/CBH (x-axis) for single-layered optically thin (Fig. 9(a)) ice cloud, single-layered optically thick (Fig. 9(b)) ice cloud, and multi-layer (Fig. 9(c)) cloud."

[27 pp.] A caption of Fig. 9: "Joint histograms of three cloud categories; (a) single-layered optically thin ice clouds, (b) optically thick ice clouds, and (c) multi-layer clouds during August 2015."

[32 pp.] A caption of Table 4, and categories in Table 4: "Comparison of max(Hc) (min(Hc)) to the CALIOP CTH (CALIOP CBH) for all cloud pixels and three cloud regimes; single-layered optically thin ice clouds, optically thick ice clouds and multi-layered clouds for August, 2015."

(4) line 18: "become larger" – Cite how large here.

The range of biases of single-layer optically thick ice clouds and multi-layered clouds are specified as follows.

[line 19–20] "For optically thick and multi-layered clouds, the biases of the estimated cloud heights from the cloud top/base become larger (0.30/–1.71 km, 1.41/–4.64 km)."

(5) Section 1 Introduction – In general, I think it would be helpful to the reader if the authors put the need for their scheme into better perspective. For example, it would be helpful to know from the outset under what circumstances non-window schemes, such as $CO_2$ -slicing, can and cannot be used for the same purpose. Likewise, is the scheme suggested in the current study expected to provide an advantage over $CO_2$ -slicing (a) because there an issue of availability of appropriate channels for implementing $CO_2$ -slicing with wide enough spatial and temporal coverage, (b) because there is some issue of reliability with the $CO_2$ -slicing technique, (c) because $CO_2$ -slicing suffers from similar biases in that the cloud is assumed to be at a single altitude and to possess a single temperature, or (d) because the current scheme is just simpler? See also my comment number 14.

Response: We revised introduction as you suggested. [lines 25–35] "Satellite sensors provide data daily that are essential for determining global cloud properties, including cloud height/pressure/temperature, thermodynamic phase (ice or liquid water), cloud optical thickness, and effective particle size. These variables are essential for understanding the net radiation of the earth and the impact of clouds (L'Ecuyer et al. 2019). In particular, cloud heights at the top and base levels are necessary to determine upwelling and downwelling infrared (IR) radiation (Slingo and Slingo, 1988; Baker, 1997; Harrop and Hartmann; 2012). Additionally, cloud heights are used to derive atmospheric motion vectors that are important for most global data-assimilation systems (Bouttier and Kelly, 2001) affecting the accuracy of the global model forecast (Lee and Song, 2018). However, in most operational retrievals of cloud properties, only a single cloud height is inferred for a given pixel, or field of view. The goal of this study is to develop an algorithm to infer cloud height boundaries for semi-transparent ice clouds using only IR measurements for its applicability of global data regardless of solar illumination. Where this study could provide the most benefit is for the case where an ice

cloud is geometrically thick but optically thin."

[lines 61–73] "There is a retrieval approach to infer optically thin cloud-top pressure that uses multiple IR absorption bands within the 15-$\mu$m CO2 band (e.g., Menzel et al. 2008; Baum et al. 2012), called the CO2 slicing method. These 15-$\mu$m CO2 band channels are available on the Terra/Aqua MODIS imagers, the HIRS sounders, and with any hyperspectral IR sounder (IASI, CrIS, AIRS). MODIS is the only imager where multiple 15-$\mu$m CO2 channels are available. Zhang and Menzel (2002) showed improvement of the retrieval of ice cloud height when they take into account spectral cloud emissivity that has some sensitivity to the cloud microphysics. As the goal of our work is to develop a reliable method for inferring ice cloud height from geostationary data, we are limiting this study to the use of the relevant IR channels, i.e., measurements at 11- 12-, and 13.3-$\mu$m. To complement the use of IR window channels, the addition of a single IR absorption channel, such as one within the broad 15-$\mu$m CO2 band, has been shown to improve the inference of cirrus cloud temperature (Heidinger et al., 2010). Their study shows how adding a single IR absorption channel at 13.3 $\mu$m to the IR 11- and 12-$\mu$m window channels decreases the solution space in an optimal estimation retrieval approach and leads to closer comparisons in cloud height/temperature with CALIPSO/CALIOP cloud products."

(6) lines 77-78: Briefly mention what scheme is used to retrieve cloud emissivity in the C6 MYD06 product.

Response: We mentioned briefly explanation how cloud emissivity for each band was retrieved in the C6 MYD06 product, as follows.

[lines 253–257] "Here we use the cloud emissivity values at 11 and 12-$\mu$m for each ice cloud pixel provided in MYD06, for which the Scientific Data Set (SDS) names are 'cloud_emiss11_1km' and 'cloud_emiss12_1km'. The cloud emissivity for a single band is obtained by the following equation: $e\_c=(I\_obs-I\_clr)/(I\_ac+T\_ac B(T\_c)-I\_clr)$. (7)"

(7) line 115: "a parameterization is adopted" – I believe that at these wavelengths, where scattering (by molecules) is negligible, $\exp(-\kappa z/\mu)$ is considered an accurate expression for the transmissivity based on the Beer-Lambert law and not a parameterization.

Response: The text "a parameterization is adopted" has been removed and revised the sentence as below.

[lines 157–158] To relate the effective emissivity between two channels, Inoue uses the relation of the cirrus emissivity to the optical thickness.

(8) line 118: "the quantity $\kappa z/\mu$ is called the optical thickness..." – I believe that $\kappa z$ is considered to be the optical thickness, rather than $\kappa z/\mu$.

Response: You are correct. The sentence now states; [line 160] "..the quantity $\kappa z$ is called the optical thickness and is also wavelength dependent."

(9) line 139: Tc is already in bold font, but it would be a good idea to emphasize in the text that this is a vector of possible values of cloud temperature rather than a single value.

Response: We emphasize Tc is a vector, not a single value in the text.

[lines 186–188] "The cloud layer temperature ranges, Tc, are estimated as a vector of possible Tc values given a range of the ec and △ec (hereafter, ec and △ec) such as ec = [e_cˆ1,e_cˆ2,âŃŕ,e_cˆn] and △ec =[△e_cˆ1,△e_cˆ2,âŃŕ,ãĂŰ△eãĂŮ_cˆn ] as shown in Fig. 2(b)."

(10) lines 151-152: In other words, one does not need to assume a ratio of cloud optical depths between the two channels, such as the ratio 1.08 in Equation 5, or is there some similar implicit assumption?

Response: Your interpretation is correct, and there is no implicit assumption either. To clarify the meaning of the sentence on lines 151-152, we revised them as the sentences

on lines 193-197.

[lines 203-207] "The third step is to find Tc values that satisfy the three equations, i.e., Eq. (2) at 11 $\mu$m, Eq. (2) at 12 $\mu$m, and the equation for cloud emissivity differences (Eq. (4)) between 11 and 12 $\mu$m with constraints in ec|11 and $\Delta$ec|11,12. That is, the last equation among the three equations in our method is different from Inoue's method (Eq. (5)) where e_c |_11=e_c |_12+ ãĂŰ$\Delta$eãĂŮ_c |_(11,12). (6)"

(11) line 155: "we obtain two T c values..." – Actually, a list of possible Tc values, including the minimum and maximum possible values, is obtained, correct?

Response: You definitely understand our intention. We clarify the sentence as below.

[lines 209–211] "That is, we obtain two Tc values as the minimum and maximum temperatures that an ice cloud pixel can have, corresponding to min/max($\Delta$ec|11,12)."

(12) lines 156-157: "... by a dynamical lapse rate..." – Does this mean that it is assumed that Tc varies within the cloud layer, or does the lapse rate only apply to the atmospheric layers outside of the cloud layer?

Response: We used the expression of 'a dynamical lapse rate' for an antonym of 'a fixed lapse rate' in the text. We added how we calculated a dynamical lapse rate in this study.

[lines 212-213] "The dynamical lapse rate on each grid is calculated from differences in temperatures between 200 and 400 hPa per differences in heights between 200 and 400 hPa."

(13) line 158: "any cloud height is not allowed..." – "no cloud height is allowed..."

Response: Corrected as noted.

[lines 214] "In this study, no cloud height is not allowed to be higher than tropopause, which is provided in the GFS NWP model product."

(14) lines 163-166: "In fact, ... day and night." – These sentences belong in the Introduction. Refer also to my comment number 5.

Response: These lines are changed in the introduction as below.

[lines 32–34] "The goal of this study is to develop an algorithm to infer cloud height boundaries for semi-transparent ice clouds using only IR measurements for its applicability of global data regardless of solar illumination."

(15) lines 97-98: "... based on an empirical relationship..." – Does this mean that the authors' Equation 4 is not used, or is the empirical relationship related to Equation 4?

Response: We removed this expression that would give confusions to readers.

(16) lines 264-266: "It is interesting that..." – I think that this should be emphasized better. It is not an interesting side note but an impressive demonstration of the concept suggested in this paper.

Response: We changed the word "interesting" to "remarkable" and added the sentence as below.

[lines 314–316] "It is remarkable that the max(Hc) corresponding to uncertainties of cloud emissivity tends to occur at or slightly above the cloud top as indicated by CALIPSO, higher than the EEL and MODIS CTH. The max(Hc) on the cloud edges and the edges of the eye of the Goni varied from the base of cloud mask and tropopause height."

[lines 317–319] "These results show the feasibility of inferring single-layered ice cloud boundaries from spectral cloud emissvity and its uncertainties by IR measurements."

(17) line 266: "... and the eye of the Goni..." – It does not seem that there are data points right in the eye, only at the edges of the eye, correct? Also "scattered from" is not the best wording here and throughout the discussion.

Response: we changed "the eye of the Goni" to "surrounding the eye of Goni", and

"scattered" to "varied"

[lines 319–321] "The max/min(Hc) on the cloud edges and the edges of surrounding the eye of the Goni have relatively large biases from the top/base of the cloud."

(18) lines 267-269: Again, I think the success of the authors' method should be emphasized better here. These sentences explain the difference between the results of their method and the other data in the regions of the cloud edges and the eye of the hurricane. However, given that the left side of the image appears to contain multiple cloud layers that are likely moving, the fact that the authors' results and the CALIOP VFM data exhibit similar variation and similar values near the tropopause actually demonstrates a rather decent qualitative correspondence between the two.

Response: We changed some expressions and added the sentence to emphasize our results as below.

[lines 313–322] "Note that the max(Hc) (blue circles) is close to the top of clouds except in the region of cloud edges and the eye of Goni. Bias between the cloud top and the max(Hc) is 0.46 km, that is –4.5 K in the aspect of temperature. It is remarkable that the max(Hc) corresponding to uncertainties of cloud emissivity tends to occur at or slightly above the cloud top as indicated by CALIPSO, higher than the EEL and MODIS CTH. The height of the min(Hc) (green circles) also follows the base of the cloud layer with a bias of -1.58 km (10.6 K in temperature), slightly lower than EEL and MODIS CTH. These results show the feasibility of inferring single-layered ice cloud boundaries from spectral cloud emissvity and its uncertainties by IR measurements. The max/min(Hc) on the cloud edges and the edges of surrounding the eye of the Goni have relatively large biases from the top/base of the cloud. Those regions show relatively large STD(Iobs|11) and small COT and contain multiple clouds. To sum up, our resulting cloud heights corresponding to cloud emissivity uncertainties are likely to exhibit similar variations to the CALIOP VFM, except the cloud edges and multiple cloud regions."

(19) lines 278-279: Once again, I think the success of the authors' method should be emphasized better here. The bias for min( Hc) from the cloud base is larger than that of optically thin clouds, but it is still better than the EEL, which is what would have been predicted.

Response: We changed the sentence as below.

[lines 331-332] "The bias for min(Hc) from the cloud base is larger than that of optically thin clouds, –2.69 km (19.4 K), but the min(Hc) still exhibit similar variation to CALIOP VFM."

(20) line 315: "... minimum value" – Was "maximum value" intended here?

Response: Good catch. We correct the sentence which you pointed out.

[lines 373–375) "This implies that maximum value of cloud height ranges corresponding to ec and ∆ec are close to the cloud top for single-layer clouds as determined from CALIOP."

(21) Figures 6-8: The orange contours are barely visible. Perhaps they are not necessary to include. Also, a closing parenthesis seems to be missing from the end of each of the three figure captions.

Response: Corrected as you suggested from Fig. 6–Fig. 7 as below. However, we keep the Fig. 8 as the previous version, since there are some water cloud pixels as the multi-layered clouds.

Please also note the supplement to this comment:
https://www.atmos-meas-tech-discuss.net/amt-2019-148/amt-2019-148-AC2-supplement.pdf

**Supplement:**

**Use of spectral cloud emissivities and their related uncertainties to infer ice cloud boundaries: Methodology and assessment using CALIPSO cloud products**

Hye-Sil Kim[1], Bryan A. Baum[2], and Yong-Sang Choi[1]

[1]Department of Climate and Energy Systems Engineering, Ewha Womans University, Seoul, Korea
[2]Science and Technology Corporation, Madison, Wisconsin, USA

*Correspondence to*: Yong-Sang Choi (ysc@ewha.ac.kr)

**Abstract.** Satellite imager-based operational cloud property retrievals generally assume that a cloudy pixel can be treated as being plane-parallel with horizontally homogeneous properties. This assumption can lead to high uncertainties in cloud heights, particularly for the case of optically thin, but geometrically thick, clouds composed of ice particles. This study demonstrates that ice cloud emissivity uncertainties can be used to provide a reasonable range of ice cloud layer boundaries, i.e., the minimum to maximum heights. Here ice cloud emissivity uncertainties are obtained for three IR channels centered at 11, 12, and 13.3 μm. The range of cloud emissivities is used to infer a range of ice cloud temperature/heights, rather than a single value per pixel as provided by operational cloud retrievals. Our methodology is tested using MODIS observations over the western North Pacific Ocean during August 2015. We estimate minimum/maximum heights for three cloud regimes, i.e., single-layered optically thin ice clouds, and single-layered optically thick ice clouds, and multi-layered clouds. Our results are assessed through comparison with CALIOP Version 4 cloud products for a total of 11873 pixels. The cloud boundary heights for single-layered optically thin clouds show good agreement with those from CALIOP; biases for maximum (minimum) heights versus the cloud top (base) heights of CALIOP are 0.13 km (–1.01 km). For optically thick and multi-layered clouds, the biases of the estimated cloud heights from the cloud top/base become larger (0.30/–1.71 km, 1.41/–4.64 km). The vertically resolved boundaries for ice clouds can contribute new information for data assimilation efforts for weather prediction and radiation budget studies. Our method is applicable to measurements provided by most geostationary weather satellites including the GK-2A advanced multi-channel infrared imager.

**1 Introduction**

Satellite sensors provide data daily that are essential for determining global cloud properties, including cloud height/pressure/temperature, thermodynamic phase (ice or liquid water), cloud optical thickness, and effective particle size. These variables are essential for understanding the net radiation of the earth and the impact of clouds (L'Ecuyer et al. 2019). In particular, cloud heights at the top and base levels are necessary to determine upwelling and downwelling infrared (IR) radiation (Slingo and Slingo, 1988; Baker, 1997; Harrop and Hartmann; 2012). Additionally, cloud heights are used to derive

30     atmospheric motion vectors that are important for most global data-assimilation systems (Bouttier and Kelly, 2001) affecting the accuracy of the global model forecast (Lee and Song, 2018). However, in most operational retrievals of cloud properties, only a single cloud height is inferred for a given pixel, or field of view. The goal of this study is to develop an algorithm to infer cloud height boundaries for semi-transparent ice clouds using only IR measurements for its applicability of global data regardless of solar illumination. Where this study could provide the most benefit is for the case where an ice cloud is

35     geometrically thick but optically thin.

[revised manuscript text omitted]

The paper is organized as follows. Section 2 describes the data used in this study. Section 3 presents the methodology and the generation of the relevant look-up tables (LUTs) for the radiances and brightness temperatures used in our analyses. Section 4 provides results for the western North Pacific Ocean during August 2015, and comparisons with CALIOP. Section 5 discusses the results and Section 6 summarizes this paper.

**2 Data**

**2.1 Study domain**

The study domain is the western North Pacific Ocean (0ºN –30ºN, 120ºE–170ºE) during three months of August from 2013–2015. Two of these months (August 2013 and August 2014) are used for generating the LUTs, while the month of August in 2015 is used for testing and validating the current algorithm. The reason for restriction of the study domain is to obtain a clear relationship between radiances/brightness temperatures and spectral cloud emissivity. In the western North Pacific Ocean, the ice clouds can be generated from diverse meteorological conditions including frequent typhoons.

**2.2 Aqua/Moderate Resolution Imaging Spectroradiometer (MODIS)**

The MODIS is a 36-channel whisk-broom scanning radiometer on the NASA Earth Observing System Terra and Aqua platforms. The Aqua platform is in a daytime ascending orbit at 1330 LST. The MODIS sensor has four focal planes that cover the spectral range 0.42–14.24 µm. The longwave bands are calibrated with an onboard blackbody. Table 1 shows the Aqua MODIS products used in this study; these products include the Collection 6 1-km Level-1b radiance data (MYD021KM), geolocation data (MYD03), and the cloud properties at 1-km resolution (MYD06). In this study, the radiances and brightness temperatures at 11, 12, and 13.3 µm (channels 31, 32, and 33, respectively) are taken from the C6 MYD021KM data. Latitude/longitude information for each granule is from C6 MYD03. The C6 MYD06 product provides cloud emissivity values in the IR window (8.5, 11, and 12 µm) and also cloud top height (CTH), all at 1-km spatial resolution; these parameters were not included in earlier collections (Menzel et al., 2008; Baum et al., 2012). The cloud emissivities at 11 and 12 µm are used in this study.

**2.3 CALIPSO/CALIOP**

The CALIPSO satellite platform carries several instruments, among which is a near-nadir-viewing lidar called CALIOP (Winker et al. 2007, 2009). Originally, CALIPSO flew in formation with NASA's Earth Observing System Aqua platform since 2006 and was part of the A-Train suite of sensors. At the time of this writing, it is no longer part of the A-Train but flies in formation with CloudSat in a lower orbit. CALIOP takes data at 532 and 1064 nm. The CALIOP 532-nm channel also measures the linear polarization state of the lidar returns. The depolarization ratio contains information about aerosol and cloud properties. This study uses CALIPSO Version 4 products that were released in November 2016. With the updated radiometric calibration at 532 and 1064 nm (Getzewich et al., 2018; Vaughan et al., 2019), cloud products such as cloud-aerosol discrimination and extinction coefficients show significant improvement relative to previous versions (Young et al., 2018; Liu et al., 2019). CALIPSO products are used to validate our retrievals, including CAL_L1D_L2_VFM-Standard-V4 which provides cloud vertical features, CAL_LID_L2_05kmCPro-Standard-V4 and CAL_LID_L2_05kmCLay-Standard-V4 which provide cloud top and base temperature (height), extinction coefficients and temperature profiles (Table 1).

**2.4 Numerical weather model product**

The Global Forecast System (GFS) model is produced by the National Centers for Environmental Prediction (NCEP) of the National Oceanic and Atmospheric Administration (NOAA) (Moorthi et al. 2001). GFS provides global NWP model outputs at 0.5º resolution at 3-hour forecast intervals every 6 hours that are available online (https://www.ncdc.noaa.gov/data-access/model-data/model-datasets/global-forcast-system-gfs). We use two variables from the NWP products, temperature profiles and geopotential heights, with cloud heights provided for 26 isobaric layers that are related to cloud temperatures. These data are used for the conversion of cloud temperatures to cloud heights. The NWP fields are remapped to the resolution of satellite imagery by linear interpolation. We use the NWP products that are closest in time to the satellite observations.

**2.5 Clear-sky maps generated from MODIS**

The MODIS pixels identified as being clear-sky are used to generate a gridded clear-sky map, which is another ancillary product required for our method. To simplify the generation of this map, the MODIS data with 1km resolution are converted to 5 km resolution. Monthly composites of clear-sky radiances ($I_{clr}$) at 0.1°×0.1° resolution are generated by choosing the maximum value among radiances for three months of August (2013–2015) in each 0.1°×0.1° grid box. To confirm the availability of the generated $I_{clr}$, we present the spatial distribution of $I_{clr}$ at 11 μm ($I_{clr}|_{11}$, Fig. 1(a)), from 8 to 11 W m$^{-2}$ μm$^{-1}$ sr$^{-1}$. The largest $I_{clr}|_{11}$ values are shown over the northwestern region of the domain, whereas the smallest $I_{clr}|_{11}$ values are shown over the southeastern region of the domain. The pattern of $I_{clr}|_{11}$ is similar to the spatial distribution of the monthly average of sea surface temperature in 2015 (https://bobtisdale.wordpress.com/2015/09/08/august-2015-sea-surface-temperature-sst-anomaly-update/). Also, we show the spatial distribution of spatial distribution of differences of $I_{clr}|_{11}$ from $I_{clr}|_{12}$ in Fig. 1(a), examining the reliability of the generated $I_{clr}|_{12}$. Note that the differences of $I_{clr}|_{11}$ and $I_{clr}|_{12}$ are positive over the domain, because water vapor absorption is stronger at 12 μm than at 11 μm. Large differences are shown in the western region, near the Philippines (green-colored contours in Fig. 1).

**3 Methodology**

**3.1 Cloud retrieval algorithm**

The basis for the retrieval algorithm is provided in Inoue (1985). Figure 2(a) shows the plane parallel homogeneous cloud model with no scattering. The ice cloud layer at a given height has a corresponding ice cloud temperature ($T_c$) and an associated cloud emissivity ($e_c$). The observed upwelling radiance ($I_{obs}$) at the cloud top is composed of two terms: the first depending on the upwelling clear-sky radiance ($I_{clr}$) at the cloud base and the other depending on the radiance ($B(T_c)$) computed for a cloud emitting as a blackbody:

$$I_{obs} = (1 - e_c)I_{clr} + e_c B(T_c), \tag{1}$$

where B($T_c$) is the Planck emission for a cloud computed at $T_c$ (Liou, 2002). All terms in Eq. (1) are wavelength dependent except for the $T_c$. $I_{obs}$ is determined from the satellite measurements, and $I_{clr}$ can be found from clear-sky conditions in the imagery or computed by a radiative transfer model given a set of atmospheric profiles of temperature, humidity, and trace gases. However, $e_c$ and $T_c$ are unknown.

Eq. (1) can be rearranged to solve for the emissivity:

$$e_c = (I_{obs} - I_{clr})/(B(T_c) - I_{clr}). \tag{2}$$

One can relate two channels by taking a ratio of the radiances, similar to that of the $CO_2$ slicing method (e.g., Menzel et al. 2008) and assuming that the emissivity between two channels spaced closely in wavelength are the same. However, Zhang

and Menzel (2002) showed improvement of the retrieval of ice cloud pressure by accounting for differences in the spectral

155    cloud emissivity.

Inoue (1985) discusses the range of uncertainties in both $T_c$ and $e_c$ and further suggests that use of multiple IR channels can reduce the uncertainties. To relate the effective emissivity between two channels, Inoue uses the relation of the cirrus emissivity to the optical thickness. The $e_c$ is a function of the absorption coefficient ($\kappa$) and the cloud thickness ($z$),

$$e_c = 1 - exp^{-\kappa z/\mu} \ . \tag{3}$$

160    The term $\mu$ in Eq. (3) is a cosine of the viewing zenith angle; the quantity $\kappa z$ is called the optical thickness and is also wavelength dependent. Given a value for $e_c$, the $T_c$ can be obtained by Eq. (2). The estimate of $e_c$ from an IR measurement will have inherent uncertainties due to the diversity of ice particle size distributions (i.e., cloud microphysics), sensor calibration, and in the cloud vertical inhomogeneity.

Another way to constrain these uncertainties is by using multiple IR channel measurements, specifically the spectral

165    emissivity differences between two IR window channels ($\Delta e_c$). We can express the $\Delta e_c$ between two IR channels by:

$$\Delta e_c = exp^{-\frac{\kappa' z}{\mu}} - exp^{-\frac{\kappa z}{\mu}}. \tag{4}$$

In Eq. (4), $\kappa'$ is the absorption coefficient at 'another' IR window channel. That is, the $\Delta e_c$ is determined by $(\kappa - \kappa')/z$ which depends on the cloud particle size and cloud thickness (Kikuchi et al., 2006). Many studies have adopted this, or a similar, approach to apply the representative relations of spectral cloud emissivity relying on cloud types to retrieve the $T_c$ (e.g., Inoue,

170    1985; Parol et al., 1991; Giraud et al, 1997; Cooper et al, 2003; Heidinger and Pavolonis, 2009).

For the case of two IR channels, Inoue (1985) formulated the retrieval of the cirrus cloud temperature and effective emissivity by setting up three equations with three unknowns (specifically referring to Inoue's equations 5, 6, and 7): Two equations are same as Eq. (2) at 11 and 12 μm in this paper, and the last equation is as follows.

$$e_c|_{12} = 1 - (1 - e_c|_{11})^{1.08}, \tag{5}$$

175    where $e_c|_{11}$ and $e_c|_{12}$ represent cloud emissivity for 11 and 12 μm, respectively. In Inoue (1985), the extinction coefficient ratio between the 11- and 12-μm channels is set to a constant value of 1.08. The cloud temperature is determined by assuming a cloud emissivity at one wavelength, calculating the emissivity at the other wavelength, and modifying the emissivities until a consistent cloud temperature is found for both wavelengths. The initial assumed 11-μm cloud emissivity begins with a value of 0 and increases by a value of 0.01 until $T_c$ converges.

180    Inoue's (1985) approach for developing the spectral cloud emissivity relationship improved the accuracy of the cirrus temperature retrievals. More recent studies explored the extinction coefficient ratio between the 11 and 12-μm channels for various cloud types (Parol et al., 1991; Duda and Spinhirne, 1996; Cooper et al., 2003). Heidinger et al. (2009) uses an optimal

estimation method that employs extinction coefficient ratios using pairs of the 8.6, 11, 12, and 13-µm channels to infer cloud heights for GOES-16/17.

185     In this study, we apply a range of spectral cloud emissivity values to infer cloud temperatures rather than an optimum value. In our approach, the cloud is considered as a number of plane parallel homogeneous cloud layers. The cloud layer temperature ranges, $\mathbf{T_c}$, are estimated as a vector of possible $T_c$ values given a range of the $e_c$ and $\Delta e_c$ (hereafter, $\mathbf{e_c}$ and $\Delta \mathbf{e_c}$) such as $\mathbf{e_c} = [e_c^1, e_c^2, \cdots, e_c^n]$ and $\Delta \mathbf{e_c} = [\Delta e_c^1, \Delta e_c^2, \cdots, \Delta e_c^n]$ as shown in Fig. 2(b). The $\mathbf{e_c}$ and $\Delta \mathbf{e_c}$ in Fig. 2(b) describes a range of possible spectral cloud emissivity values that can simulate the measured channel radiances. Thus, this study aims to produce $\mathbf{T_c}$ given

190     the $\mathbf{e_c}$ and $\Delta \mathbf{e_c}$, and to examine how closely the retrieved $\mathbf{T_c}$ are to the actual vertical cloud structure.

The differences between this study and Inoue (1985) are summarized as follows.

1. Constraints in the iteration range for cloud emissivity are provided in look-up tables (LUTs) discussed in the next section, as opposed to considering the full range of possible values from 0 to 1.

2. Emissivity differences ($\Delta \mathbf{e_c}$) are used, rather than a single value for the extinction coefficient ratio between two infrared

195     channels.

3. Given the range of emissivity differences ($\Delta \mathbf{e_c}$ provided in LUTs), we obtain a range of $\mathbf{T_c}$ (and hence a range of cloud heights, $\mathbf{H_c}$) that can be compared to CALIPSO products.

The first step in the current method (Fig. 3) is to constrain 11-µm cloud emissivity ranges ($\mathbf{e_c}|_{11}$) that an ice cloud pixel can have based on the brightness temperatures. To obtain a reasonable $\mathbf{e_c}|_{11}$ boundary corresponding to the ice cloud microphysical

200     properties, the LUTs are generated to provide $\mathbf{e_c}|_{11}$ ranges characterized by  brightness temperature (BT) for 11 µm (BT$|_{11}$), BT differences (or BTD) between 11 and 13 µm (BTD$|_{11, 13}$) and between 11 and 12 µm (BTD$|_{11, 12}$) (the light gray box in Fig. 3).

The second step is to constrain cloud emissivity differences between 11 and 12 µm for an ice cloud pixel, $\Delta \mathbf{e_c}|_{11,12}$ that are also provided in LUTs (the dark gray box in Fig. 3) with identical input parameters as in the first step. The third step is to find $\mathbf{T_c}$ values satisfying the three equations, i.e., Eq. (2) at 11 µm, Eq. (2) at 12 µm, and the equation for cloud emissivity

205     differences (Eq. (4)) between 11 and 12 µm with constraints in $\mathbf{e_c}|_{11}$ and $\Delta \mathbf{e_c}|_{11,12}$.  That is, the last equation among the three equations in our method is different from Inoue's method (Eq. (5)) where

$$e_c|_{11} = e_c|_{12} + \Delta e_c|_{11,12}. \tag{6}$$

The initial assumed 11-µm cloud emissivity begins with a value of min($\mathbf{e_c}|_{11}$) and increases by a value of 0.01 until $T_c$ converges. Notice that the $T_c$ value, an element of available ice cloud temperatures set as $\mathbf{T_c}$, depends on $\Delta \mathbf{e_c}|_{11,12}$ in Eq. (4). That is, we

210     obtain two $T_c$ values as the minimum and maximum temperatures that an ice cloud pixel can have, corresponding to min/max($\Delta \mathbf{e_c}|_{11,12}$). Finally, we estimate cloud height ranges, $\mathbf{H_c}$, relating to min/max($\mathbf{T_c}$) using a dynamical lapse rate calculated from GFS NWP temperature profiles provided for 26 isobar layers. The dynamical lapse rate on each grid is calculated from differences in temperatures between 200 and 400 hPa per differences in heights between 200 and 400 hPa. In this study, no cloud heights are not allowed to be higher than tropopause, which is provided in the GFS NWP model product.

**3.2 Generation of look-up tables (LUTs)**

For our method, relevant information for the western North Pacific Ocean is stored in look-up tables (LUTs). The LUTs include the min/max($e_c$) and min/max($\Delta e_c$) values for three indices: $BTD|_{11,13}$, $BTD|_{11,12}$, and $BT|_{11}$. The reason for selecting these three indices is that they are linked with cloud optical thickness, cloud effective radius, and cloud temperatures, respectively. Both solar and infrared radiances have been used to investigate cloud microphysics using passive satellite measurements (e.g., Freud et al., 2008; Lensky and Rosenfeld, 2006; Martins et al., 2011). A primary benefit of using IR measurements is that the ice cloud temperature and emissivity do not depend on solar illumination, so the cloud properties are consistent between day and night.

First, the $BTD|_{11,13}$ is sensitive to the presence of mid- to high-level clouds and the cloud height. While both the 12- and 13.3-μm measurements are both affected by $CO_2$ absorption, the 12-μm channel is at the wing of the broad 15-μm $CO_2$ band and has less $CO_2$ absorption than the 13.3-μm channel. Additionally, the peak of weighting function for the 13.3-μm channel is in the vicinity of 700-800 hPa so that the observed radiance at 13.3 μm represents the lower tropospheric temperature. Thus, the BT at 13.3 μm is generally colder than that of the two other IR window channels. The $BTD|_{11,13}$ is larger for clear-sky pixels than for ice clouds, but $BTD|_{11,13}$ depends on degree of cloud opacity. The $BTD|_{11,13}$ has been applied by Mecikalski and Bedka (2006) to monitor changes in cloud thickness and height for signals of convective initiation.

Second, the $BTD|_{11,12}$ depends in part on the microphysics and cloud opacity, i.e., the number and distribution of the ice particles; the imaginary part of the refractive index for ice varies in the IR region under study. The $BTD|_{11,12}$ has been used to identify cloud type (Inoue, 1985; Pavolonis and Heidinger, 2004; Pavolonis et al.,2005). Prata (1989) used the $BTD|_{11,12}$ to discern volcanic ash from non-volcanic absorbing aerosols. Recently, adding BTD from 8.6 and 11 μm, the $BTD|_{11,12}$ is also applied to infer cloud phase (Strabala et al.,1994; Baum et al., 2000, 2012).

Finally, $BT|_{11}$ values can provide cloud height information, at least for optically thick clouds including low-level clouds. For optically thick clouds, the $BT|_{11}$ values approximate the actual cloud temperature, since at 11μm the primary absorber is water vapor and there is generally little absorption above high-level ice clouds. As noted earlier, the $BT|_{11}$ for optically thin clouds includes a contribution from upwelling radiances from the surface and lower atmosphere.

The LUTs are compiled for $e_c$ and $\Delta e_c$ by three input parameters, i.e., $BTD|_{11,13}$, $BTD|_{11,12}$, and $BT|_{11}$ from information in the C6 MODIS products. Data used in generating our LUTs are summarized in Table 1. The first step is to collect all ice cloud radiances at 11, 12, and 13.3 μm from MYD021KM over the western North Pacific Ocean during the recurring period of the August 2013 and 2014. Ice cloud pixels are identified by the MODIS IR cloud thermodynamic phase product in MYD06 (Baum et al. 2012) and where the pixels have a cloud top temperature ≤ 260 K. The spatial and temporal domain is restricted to obtain a clear relationship between spectral cloud emissivity and three IR parameters for the case study analyses that will be presented in Section 4.

The second step is to categorize the ensemble of ice cloud pixels by three parameters, $BTD|_{11,13}$, $BTD|_{11,12}$, and $BT|_{11}$. The collected cloud pixels are separated into cloud types linked with cloud microphysical properties. We convert radiances centered

at 11, 12, and 13.3 μm to BT by the inverse Planck's function and then calculate BTD|$_{11,13}$, BTD|$_{11,12}$, and BT|$_{11}$ for each pixel. Subsequently the ice cloud pixels are sorted into range bins defined for the three parameters as follows: BT|$_{11}$ values in a range from 190 K to 290 K in increment of 5 K; BTD|$_{11,13}$ values in a range from –2 K to 30 K in increments of 2 K; and BTD|$_{11,12}$ values ranging from –1 K to 10 K in increments of 0.5 K (Table 2). For example, the first category is 190 K ≤ BT|$_{11}$ < 195 K, –2 ≤ BTD|$_{11,13}$ < 0, and –1 ≤ BTD|$_{11,12}$ < –0.5.

The final step is to find the possible ranges of $e_c$ and $\Delta e_c$ in each of the bins of BTD|$_{11,13}$, BTD|$_{11,12}$, and BT|$_{11}$. Here we use the cloud emissivity values at 11 and 12-μm for each ice cloud pixel provided in MYD06, for which the Scientific Data Set (SDS) names are 'cloud_emiss11_1km' and 'cloud_emiss12_1km'. The cloud emissivity for a single band is obtained by the following equation:

$$e_c = (I_{obs} - I_{clr})/(I_{ac} + T_{ac}B(T_c) - I_{clr}). \tag{7}$$

In Eq. (7), $T_{ac}$ and $I_{ac}$ are the above-cloud transmittance and the above-cloud emission (Baum et al., 2012), which are additional terms compared to the definition of the cloud emissivity in the infrared window regions in this paper (Eq. (2)). In spite of different definition of Eq. (7) from the Eq. (2), we use this cloud emissivity data since there the differences are small from the two different equations in the infrared window region. Note that the cloud emissivity data from C6 MYD06 are retrieved under the assumption of the single-layered cloud. Here the possible ranges of $e_c$ and $\Delta e_c$ are determined as the min/max($e_c$) and ($\Delta e_c$) among cloud emissivity values allocated by the bins of three parameters. To exclude extreme values, the min/max($e_c$) and ($\Delta e_c$) are defined as the 2nd /98th percentiles of the $e_c$ and $\Delta e_c$ distributions when there are at least 5,000 pixels available for a given bin. When there are between 500 and 5000 pixels, the 5th /95th percentiles are chosen as the min/max($e_c$) and ($\Delta e_c$). In the rare case when there are between only 200 and 500 pixels, the 10th /90th percentiles are used. Any case with fewer than 200 ice cloud pixels is not included in the LUTs.

Fig. 4 shows examples of LUT values for $e_c$ belonging to the specific category for 230 K ≤ BT|$_{11}$ < 235 K (Fig. 4(a)) and 270 K ≤ BT|$_{11}$ < 275 K (Fig. 4(b)), which imply the presence of optically thick and thin ice clouds, respectively. The minimum (the left panel) and maximum (the right panel) values of the $e_c$ are shown as colors in the space of BTD|$_{11,12}$ (x-axis) and the BTD|$_{11,13}$ (y-axis). In Fig. 4(a), the $e_c$ values range from about 0.8 to 1.1. The $e_c$ generally ranges from 0 to 1, but a non-physical $e_c$ value over 1 might occur in case of an over-shooting cloud (from strong convection that briefly enters the lower stratosphere) that has colder temperature than surrounding environment temperature (Negri, 1981; Adler et al., 1983). As for optically thin clouds, the $e_c$ values of Fig. 4(b) range from around 0.3 to 0.8. In general, $e_c$ values are low when cloudy pixels have large values of BTD|$_{11,12}$ and BTD|$_{11,13}$.

Fig. 5 shows examples of LUT values of $\Delta e_c$ for optically thick (Fig. 5(a)) and thin (Fig. 5(b)) ice clouds as shown in Fig. 4. The $\Delta e_c$ ranges from –0.12 to 0.04. The $\Delta e_c$ shows a more complex relationship with BTD|$_{11,12}$ and BTD|$_{11,13}$ than with $e_c$. It is notable that similar patterns $\Delta e_c$ are repeated on the optically thick (Fig. 5(a)) and thin ice cloud (Fig. 5(b)). One reason for this could be that $\Delta e_c$ are more sensitive to particles sizes, whereas $e_c$ values are more directly linked with cloud opacity (refer to Eq. (3) and Eq. (4)). The optically thin ice cloud cluster tends to be more sensitive to BTD|$_{11,12}$, showing larger variations of $\Delta e_c$ than the thick ice cloud cluster.

**4 Results**

The current algorithm analyses are performed over the study domain, the western North Pacific Ocean, in August 2015. Note that the typhoon 'Goni' formed on 13 August and dissipated on 30 August, 2015, and affected East Asia. Case studies involving Typhoon Goni scenes are provided in Section 4.1. Quantitative analysis and comparison of our results with CALIOP cloud products are described in the Section 4.2.

**4.1 Comparison of min/max($H_c$) with CALIPSO for three granules**

**4.1.1 A scene for single-layered optically thin ice clouds (19 August, 2015, at 0320 UTC)**

Figure 6 is a scene analysis for single-layered optically thin ice clouds for a granule at 0320 UTC on 19 August, 2015. Fig. 6(a) is a MODIS false color image that captures Tropical Cyclone Goni. Note that the image is rotated 90 degrees left to simplify comparison with CALIPSO. The heavy pink line (Fig. 6(a)) is the south-to-north CALIPSO track at the closest time to the MODIS observation time. The CALIPSO made a near-eye overpass of the cyclone. The CALIOP track measures a cross section of the cyclone, from the eyewall to the outer bands. Fig. 6(b) is a cross section from CALIOP data (Table 3) at the time of the overpass, that shows the horizontal (x-axis) and vertical (y-axis at the left side) locations of all cloud layers. The CALIOP vertical feature mask (VFM) indicates the presence of randomly-oriented ice and horizontally-oriented ice (sky-blue) in the scene. The y-axis at the right side is for two supplementary data shown as gray lines. The gray solid line is the CALIOP COT at 532 nm, for the opacity of ice clouds. The gray dashed line is the standard deviation of the MODIS $I_{obs}|_{11}$ (STD($I_{obs}|_{11}$)) on the collocated path with the CALIOP track, calculated over a $5 \times 5$ pixel array centered at each cloud pixel. The STD($I_{obs}|_{11}$) includes cloud feature information (Nair et al., 1998). For example, pixels at cloud edges or fractional clouds have relatively large STD($I_{obs}|_{11}$). The STD($I_{obs}|_{11}$) values are used to filter overcast cloud pixels. The data in Fig. 6 are primarily of single-layered ice clouds with horizontal homogeneity as demonstrated by the low value of STD($I_{obs}|_{11}$).

For comparison with CALIPSO, the min/max($T_c$) are converted to max/min($H_c$) and are shown from our method (blue/green circles) to the VFM in Fig. 6(b). Also provided is the MODIS CTH (black circles) for reference. For these comparisons, we converted temperature to height using a dynamical lapse rate from GFS NWP temperature profiles. When the cloud pixel temperature is colder than the tropopause temperature, it is changed to be that of the tropopause and is converted to the tropopause height provided by GFS NWP. The solid red line indicates where the CALIOP COT is about 0.5. This line is a reference for the position where the passive remote sensing retrievals will place the cloud (Holz et al. 2006; Wang et al., 2014), well known as the radiative emission level. The radiative emission level should be thought of more as a guideline since the matched COT values can be different depending on cloud types or algorithm methods. To determine this depth in the cloud layer, we integrated the extinction coefficient, CALIOP $Q_e$ (Table 3), from the top of the cloud downwards until the COT reached about 0.5. Hereafter, we call that layer as the effective emission layer, EEL. The enhancement of EEL at approximately 15.6°N in Fig. 6(b) is caused by an extraordinary value of $Q_e$ provided in the CALIOP V4.

Note that the max($\mathbf{H_c}$) (blue circles) is close to the top of clouds except in the region of cloud edges and the eye of Goni. Bias between the cloud top and the max($\mathbf{H_c}$) is 0.46 km, that is –4.5 K in the aspect of temperature. It is remarkable that the max($\mathbf{H_c}$) corresponding to uncertainties of cloud emissivity tends to occur at or slightly above the cloud top as indicated by CALIPSO, higher than the EEL and MODIS CTH. The height of the min($\mathbf{H_c}$) (green circles) also follows the base of the cloud layer with a bias of −1.58 km (10.6 K in temperature), slightly lower than EEL and MODIS CTH. These results show the feasibility of inferring single-layered ice cloud boundaries from spectral cloud emissvity and its uncertainties by IR measurements. The max/min($\mathbf{H_c}$) on the cloud edges and the edges of surrounding the eye of the Goni have relatively large biases from the top/base of the cloud. Those regions show relatively large STD($I_{obs}|_{11}$) and small COT and contain multiple clouds. To sum up, our resulting cloud heights corresponding to cloud emissivity uncertainties are likely to exhibit similar variations to the CALIOP VFM, except the cloud edges and multiple cloud regions.

**4.1.2 A scene for single-layered optically thick ice clouds (19 August, 2015, at 1530 UTC)**

The second case is the single-layered optically thick ice clouds (Fig. 7) at 1530 UTC on 19 August 2015. Here we show the BT$|_{11}$ image instead of RGB image (Fig. 7(a)) since this is a nighttime scene. Fig. 7(a) is also rotated 90 degrees left. For this overpass, CALIOP observed clouds farther away from the center of Goni, and inspection of the cross-section in Fig. 7(b) suggests that most of cloud pixels are optically thick with COT values higher than 5, about where the CALIOP signal attenuates, and have relatively low STD($I_{obs}|_{11}$) as indicated by the gray solid/dashed line in Fig. 7(b). In the comparison with the CALIOP VFM, the max($\mathbf{H_c}$) tends to occur at or slightly below the cloud top as indicated by CALIPSO, still higher than the EEL and MODIS CTH. The bias for the max($\mathbf{H_c}$) from the top of clouds is 2.38 km (–13.2 K), which is larger than that of optically thin ice clouds. The bias for min($\mathbf{H_c}$) from the cloud base is larger than that of optically thin clouds, –2.69 km (19.4 K), but the min($\mathbf{H_c}$) still exhibit similar variation to CALIOP VFM. The passive IR measurements have an upper COT limit as shown in earlier studies (Heidinger et al. 2009; 2010). The height boundaries from our method brackets both the CALIPSO measurements and the MODIS retrievals.

**4.1.3 A scene for multi-layered cloud (8 August, 2015, at 0520 UTC)**

The third case also involves a cross-section of Goni, but this scene is more complex in that there is evidence of both multi-layered and less homogeneous ice clouds on the southern boundary of the typhoon (Fig. 8a). Note that the STD ($I_{obs}|_{11}$) on the CALIPSO track show relatively large variances, compared to the previous two cases (Fig. 8(b)). The CALIOP COT is omitted given the high fluctuations in the values. The CALIOP vertical feature mask (VFM) indicates the presence of randomly-oriented ice and horizontally-oriented ice (sky-blue) including water (orange) cloud phase. The enhancement of EEL at around 25.7ºN in Fig. 8(b) is also caused by an extraordinary value of $Q_e$ provided in the CALIOP V4 product. In the region of 10ºN–20ºN, the max/min($\mathbf{H_c}$) in this region are often outside the boundaries of the VFM. The max($\mathbf{H_c}$) (blue circles) varied from near the second cloud layer to the top of the first cloud at the tropopause. Some pixels of the min($\mathbf{H_c}$) (green circles) values are also outside the range of the VFM. There is more than one reason causing these increased variances, including the fact that

345   the uppermost cloud layer is optically thin (over half of all pixels have COT < 1.5) and there are indications of lower cloud layers. In the region of 20ºN–30ºN, clouds on the top layer are relatively thick (on average, COT = 3.5). In that case, heights of the max($H_c$) on the multi-layer pixels tend to be close to the EEL, which is much lower than the top of clouds. This is to be expected for the case of a geometrically thick but optically thin cloud. Note that the value of the min($H_c$) on the multi-layered cloud pixels sometimes reach almost to the second cloud layer, rather than near the first layer. Further thought needs to be

350   given to these cases.

**4.2 Comparison of max/min $H_c$ with CALIPSO for August 2015**

In this section, the max/min($H_c$) is compared with the cloud top/base height (CTH/CBH) from CALIOP over the western North Pacific during August 2015. The computationally efficient method of Nagle et al. (2009) is used to collocate the simultaneous nadir observations (SNO) between two satellites. Following their approach, CALIOP is projected onto MODIS.

355   First, we qualitatively examine the max/min($H_c$) with the cloud layer vertical cross-section from CALIOP/MODIS matchup files (Table 3) in Fig. 6-Fig. 8. Second, we quantitatively investigate the max/min ($H_c$) for all ice clouds against CALIOP CTH/CBH during the month. The extinction coefficients profiles, cloud phase and their quality flags, and the number of cloud layers are extracted from CALIOP and used in this analysis (Table 3).

  The matchup data are filtered as follows: only ice cloud phase pixels are chosen that have the highest quality (CALIOP QC

360   for cloud phase = 1), where CALIOP COT > 1.5 and STD($I_{obs}|_{11}$) from MODIS ≤ 1, which helps to remove cloud edges and fractional clouds. The relationship is investigated between the max/min($H_c$) and CALIOP CTH/CBH for three cloud regimes; (1) single-layered optically thin ice clouds, (2) optically thick ice clouds, and (3) multi-layered clouds where the uppermost layer is optically thin cirrus. The CALIOP/MODIS matchup clouds are separated into single-layered and multi-layered cloud groups using the number of layers found (NLF) from CALIOP (Table 3). The multi-layered cloud group includes two or more

365   cloud layers, excluding single-layered clouds. Among single-layered cloud pixels, we define optically thin/thick cloud groups as CALIOP COT which is less/greater than 3.5, referring to the ISCCP cloud classification (Rossow et al., 1985; Rossow and Schifer,1999).

  Fig. 9 shows the joint histogram of the max/min($H_c$) (y-axis of left/right panels) as a function of the CALIOP CTH/CBH (x-axis) for single-layered optically thin ice cloud (Fig. 9(a)), single-layered optically thick ice cloud (Fig. 9(b)), and multi-

370   layered clouds (Fig. 9(c)). Table 4 provides all statistical quantities for Fig. 9 as correlations (corr), differences of the mean value (bias), and root mean square differences (rmsd). Additionally, all statistical quantities in terms of temperature are in the unit of K and are given in the round brackets in Table 4. For single-layered clouds, the majority of max($H_c$) values are scattered about the one-to-one line. The statistical values are corr = 0.61, bias = 0.13 km, rmsd = 0.91 for thin clouds. This implies that maximum value of cloud height ranges corresponding to $\mathbf{e_c}$ and $\Delta\mathbf{e_c}$ are close to the cloud top for single-layered clouds as

375   determined from CALIOP.

  However, the scatter is higher for optically thick clouds, with corr = 0.65, bias = 0.30 km, rmsd = 1.08 (Table 4). As for the max($H_c$) for multi-layered clouds, the majority of scatter points are on the lower right side of the one-to-one line, with corr =

0.25, bias = 1.41 km, and rmsd = 2.64. The lowest correlation and the largest bias for multi-layered clouds are shown, as expected given the assumption of the single-layered clouds in our method.

380     The comparisons of the min($\mathbf{H_c}$) (y-axis of right panels in Fig. 9) to the CALIOP CBH (x-axis) for all cloud categories show relatively large correlations, at least over 0.48. Scatter points in three joint histograms for all cloud types are parallel to the one-to-one line, but show negative biases implying higher heights than CALIOP CBT. As with the cases of the max($\mathbf{H_c}$), bias of the min($\mathbf{H_c}$) increases from single-layered optically thin ice (–1.01 km), to optically thick ice (–1.71 km) and multi-layered clouds (–4.64km).

385  **5 Discussion of Results**

The results in Figs. 6–9 show the comparisons of the ice cloud height ranges obtained based on the ice cloud emissivity uncertainties with both MODIS C6 products and vertical cross sections of clouds from CALIOP. We investigated minimum and maximum ice cloud heights for each cloud pixel for three cloud regimes during August 2015: (1) single-layered optically thin clouds, (2) optically thick ice clouds, and (3) multi-layered clouds.

390     Overall, the maximum values of the estimated ice cloud height ranges for single-layered optically thin/thick ice clouds show some skill in comparison with the cloud tops from CALIOP: corr = 0.61/0.65, bias = 0.13/0.30 km. In particular, we note that the upper height boundary for optically thin clouds derived from our method are very close to the geometric cloud tops. For multi-layered clouds, the maximum heights are occasionally much lower than the uppermost cloud layer as observed by CALIOP, showing the highest bias at 1.41 km. Higher biases are expected in our method given the assumption of single-

395  layered clouds in each pixel. Additionally, the skill of our method decreases when the upper cloud layer is composed of optically thin (having very low COT values) and fractional clouds; in some cases, the method cannot determine an emissivity range from the LUTs, which were generated for single-layered ice clouds.

The minimum heights for single-layered optically thin ice clouds reach near the base of cloud, with corr = 0.83, bias = –1.01 km. However, those for thick/multilayer, the biases became larger, at most –4.64 km. That is, the minimum heights for

400  thick clouds became much higher than the CALIOP, the observed cloud bases. This indicates that the IR method has an optical thickness limitation and is more useful for lower optical thicknesses, which has been noted previously (e.g., Heidinger et al. 2010). Even with large biases of minimum heights, it is notable that correlation coefficients between minimum heights and the cloud base for all three cloud regimes are sufficiently large, at least 0.48.

To better understand the potential biases of the current algorithm in comparison with CALIOP, we compare the mean($\mathbf{H_c}$)

405  to the mean(CALIOP $\mathbf{H_c}$) that are defined as 0.5·(max($\mathbf{H_c}$)+min($\mathbf{H_c}$)) and as 0.5·(CALIOP CTH + CALIOP CBH), respectively. Fig. 10 shows the frequency of occurrence of biases, that is, the mean(CALIOP $\mathbf{H_c}$) minus the mean($\mathbf{H_c}$), as a function of CALIOP COT for the single-layered ice clouds during August 2015. In a comparison of the MODIS cloud mask with CALIOP, Ackerman et al., (2008) noted that the cloud mask performs best at optical thicknesses above about 0.4. The lidar has a greater sensitivity to particles in a column than passive radiance measurements. Based on this consideration, we limited our results to

410  those pixels where the COT ≥ 0.5 in x-axis of Fig. 10.

Fig.10 illustrates that our resulting single-layered ice clouds boundaries are consistent with CALIOP measurements, showing slightly negative biases except the region near 'COT≤1.5'. These results suggest that our approach for applying a range of cloud emissivity values to estimate cloud boundaries has potential merit for using IR channels to produce cloud boundaries similar to those that the lidar observes, especially for optically thin but geometrically thick ice clouds which tend to have large uncertainties (Hamann et al., 2014).

The negative biases of the mean($H_c$) from CALIOP measurements are caused primarily by two factors: (1) The min($H_c$) values for all cloud regimes tend to be higher than geometric cloud base, and (2) The max($H_c$) values are sometimes slightly outside the actual cloud boundaries. Perhaps this is caused in part by the conversion of temperature to height using the NWP model product. Another source of error could be that the radiances have some amount of uncertainty that was not considered in our methodology. The notable point is that the boundary heights for optically thin cirrus (1.5<COT≤3.5) show the lowest biases.

[revised manuscript text omitted]

585   **Figure 2: The conceptual model for (a) a plane parallel homogeneous cloud layer with no scattering, characterized by cloud emissivity ($e_c$) and cloud emissivity differences between two infrared channels ($\Delta e_c$) at the cloud temperature ($T_c$) and  (b) a number of plane parallel homogeneous cloud layers (the stripes box)  with a possible range of $e_c$ and $\Delta e_c$ such as $e_c = [e_c^1, e_c^2, \cdots, e_c^n]$ and $\Delta e_c = [\Delta e_c^1, \Delta e_c^2, \cdots, \Delta e_c^n]$ corresponding to a possible range of cloud temperature, $T_c = [T_c^1, T_c^2, \cdots, T_c^n]$, where $I_{clr}$ and B are the clear-sky radiance and the Planck's function, respectively.  Arrows represent upwelling radiances.**

590

595

[revised manuscript text omitted]
| Single-layered optically thick ice cloud | NLF = 1 COT > 3.5 | 3067 | 0.65 (0.66) | 0.30 (−1.53) | 1.08 (7.12) | 0.87 (0.87) | −1.71 (13.96) | 1.92 (15.53) |
| Multi-layered cloud | NLF > 1 | 6569 | 0.25 (0.23) | 1.41 (−10.18) | 2.64 (19.53) | 0.48 (0.48) | −4.64 (31.22) | 5.95 (38.69) |

735 **Table 4. Comparison of max(H$_c$) (min(H$_c$)) to the CALIOP CTH (CALIOP CBH) for all cloud pixels and three cloud regimes; single-layered optically thin ice clouds, optically thick ice clouds and multi-layered clouds for August, 2015. Pixel numbers (count), correlation coefficients (corr) and differences of the mean values (bias), root mean square differences (rmsd) are provided. Additionally, comparison of min/max(T$_c$) to the CALIOP CTT/CBT are also shown as numbers of round brackets.**

740

---

## Author Comment (AC3) · 6 Aug 2019

General comments: This paper described a method to estimate the height of ice clouds from satellite measurements of three infrared channels. The paper is well written and includes useful information for researchers in the field of satellite remote sensing. However, it was difficult to understand how the vertical inhomogeneity of ice clouds was considered in this method. The reviewer concluded that additional explanation is necessary in the manuscript before AMT publication. Specific comments are addressed below.

We greatly appreciate your detailed comments, which we used to revise and improve

[Figure]

our paper as shown below.

(1) Page 5 line 138-140 and Fig.1: What does e_cËȨi (i=1âN ÌĄ ÌĄṛn) in Fig.1b mean? Why e_c and T_c represent inhomogeneous layer? In the reviewer's understanding, e_c (and T_c) describes a range of possible cloud emissivity (and temperature) that can simulate the measured channel radiances assuming a homogeneous cloud layer.

Response: Modified as suggested. [Lines 188 –190] "The ec and $\Delta$ec in Fig. 2(b) describes a range of possible spectral cloud emissivity values that can simulate the measured channel radiances. Thus, this study aims to produce Tc given the ec and $\Delta$ec, and to examine how closely the retrieved Tc are to the actual vertical cloud structure."

[the caption of Figure 2, 20 pp] Figure 2: The conceptual model for (a) a plane parallel homogeneous cloud layer with no scattering, characterized by cloud emissivity (ec) and cloud emissivity differences between two infrared channels ($\Delta$ec) at the cloud temperature (Tc) and (b) a number of plane parallel homogeneous cloud layers (the stripes box) with a possible range of ec and $\Delta$ec such as ec = [e_cˆ1,e_cˆ2, âŃŕ,e_cˆn] and $\Delta$ec = [ãĂŰ$\Delta$eãĂŮ_cˆ1,ãĂŰ$\Delta$eãĂŮ_cˆ2, âŃŕ,$\Delta$e_cˆn] corresponding to a possible range of cloud temperature, Tc = [T_cˆ1,T_cˆ2, âŃŕ,T_cˆn], where Iclr and B are the clear-sky radiance and the Planck's function, respectively. Arrows represent upwelling radiances.

(2) Page7 line 197-200: I suppose that the LUT for the empirical relationship between cloud emissivity and BT/BTD is a key of the proposed method. Does the author assume the dataset MYD021KM and MYD06 provide the relationship for vertically inhomogeneous cloud layer? I think that the author should express the basic idea of your approach for inhomogeneous cloud layer in the manuscript.

Response: The LUTs for the relationship between cloud emissivity and BT/BTD were constructed using MYD021KM and MYD06 products. Even MYD06 products are retrieved under the assumption of the single-layer cloud, they are useful to express uncertainties in cloud emissivity due to diverse cloud microphysical properties that are likely to exist in the vertical cloud structure. Thus, we explicitly stated that vertical inhomogeneity was not considered in the generated LUTs and also explained how those LUTs can work to infer uncertainties in cloud emissivity in the vertical cloud structure in the revised manuscript as below.

[Lines 261–267] "Note that the cloud emissivity data from C6 MYD06 are retrieved under the assumption of the single-layered cloud. Here the possible ranges of ec and $\Delta$ec are determined as the min/max(ec) and ($\Delta$ec) among cloud emissivity values allocated by the bins of three parameters. To exclude extreme values, the min/max(ec) and ($\Delta$ec) are defined as the 2nd /98th percentiles of the ec and $\Delta$ec distributions when there are at least 5,000 pixels available for a given bin. When there are between 500 and 5000 pixels, the 5th /95th percentiles are chosen as the min/max(ec) and ($\Delta$ec). In the rare case when there are between only 200 and 500 pixels, the 10th /90th percentiles are used. Any case with fewer than 200 ice cloud pixels is not included in the LUTs."

(3) Page 8 line 233-237 and Fig.5: The reviewer cloud not understand what does the author intend to show in Fig.5a and 5b. What does the region of large differences of I_(clr|11)-I_(clr|12) in Fig.5b suggest?

Response: Our intent for Figs. 5a and 5b (Fig. 1a and Fig. 1b in the revised manuscript) is (hopefully) more clear in the text and replicated as follows.

[lines 126–137] "The MODIS pixels identified as being clear-sky are used to generate a gridded clear-sky map, which is another ancillary product required for our method. To simplify the generation of this map, the MODIS data with 1km resolution are converted to 5 km resolution. Monthly composites of clear-sky radiances (Iclr) at $0.1° \times 0.1°$ resolution are generated by choosing the maximum value among radiances for three months of August (2013–2015) in each $0.1° \times 0.1°$ grid box. To confirm the availability of the generated Iclr, we present the spatial distribution of Iclr at 11

$\mu$m (Iclr|11, Fig. 1(a)), from 8 to 11 W m-2 $\mu$m-1 sr-1. The largest Iclr|11 values are shown over the northwestern region of the domain, whereas the smallest Iclr|11 values are shown over the southeastern region of the domain. The pattern of Iclr|11 is similar to the spatial distribution of the monthly average of sea surface temperature in 2015 (https://bobtisdale.wordpress.com/2015/09/08/august-2015-sea-surface-temperature-sst-anomaly-update/). Also, we show the spatial distribution of spatial distribution of differences of Iclr|11 from Iclr|12 in Fig. 1(a), examining the reliability of the generated Iclr|12. Note that the differences of Iclr|11 and Iclr|12 are positive over the domain, because water vapor absorption is stronger at 12 $\mu$m than at 11 $\mu$m. Large differences are shown in the western region, near the Philippines (green-colored contours in Fig. 1)."

(4) Page 22 Fig.6b and Page 8 Fig.8b: What is the enhancement of EEL at latitude 15.6âŮ̧ę of Fig. 6b? Similar enhancement is also appeared at latitude 25.7âŮ̧ę in Fig. 8b. Response: Added the explanation about the enhancement of EEL shown in Fig. 6b and Fig. 8b, as detailed below.

[Line 311–312] The enhancement of EEL at approximately 15.6°N in Fig. 6(b) is caused by an extraordinary value of Qe provided in the CALIOP V4. [Line 340–341] The enhancement of EEL at around 25.7°N in Fig. 8(b) is also caused by an extraordinary value of Qe provided in the CALIOP V4 product.

Please also note the supplement to this comment:
https://www.atmos-meas-tech-discuss.net/amt-2019-148/amt-2019-148-AC3-supplement.pdf
* * *

---

## Author Comment (AC4) · 6 Aug 2019

General comments) This study uses spectral cloud emissivity to derive information regarding the minimum and maximum values of cloud top height (CTH). Authors primarily use MODIS data to derive the relationship between brightness temperature (BT) or brightness temperature difference (BTD) and emissivity values to infer information of cloud top temperature (CTT), and then convert CTT into CTH. They used CALIPSO data to validate their products. Though such type of study is essential to improve our understanding regarding CTH retrieval accuracy by MODIS and other satellite sensors, this study needs more improvement to full this gap as explained in detail in the specific

comments below. The present version of the manuscript needs substantial revision. The presentation is not clear and discussion is relatively poor. The study method (Figure 2) is ambiguous. For example, what information do authors use from ice cloud pixels to determine the permissible ec?, what is the meaning of permissible ec ?, do authors use emissivity data or uncertainty in emissivity? There are a number of such confusions to the reader. Further, It is not clear how this study can address the problem of cloud vertical inhomogeneity as stated in the first line of abstract. It should be either removed or discussions are necessary to show how this study can address such problem. The discussions presented in the second half are relatively poor. For example, what are authors' view for relatively large difference in min(Hc) and CALIOP base height in Figure 9?. The English also needs to be improved.

Your detailed comments helped us to revise and improve our paper. We hope that this revised manuscript sufficiently addresses your comments and improves the clarity.

Specific comments

1. L63: Write the full form of NWP as it appears for the first time here.

Response: Done.

[lines 77–79] "The emissivities are used subsequently to estimate ranges of cloud height, which are found by converting the estimated cloud temperature ranges using a simple linear interpolation of the Numerical Weather Prediction (NWP) model products."

2. Section 2 : It is better to separate data and methodology in different sections.

Response: Done. Section 2 describes the data used in this study and Section 3 presents the methodology to infer cloud heights.

3. L95: Specify what method is used while remapping NWP fields to the resolution of satellite imaginary and interpolating to the time corresponding to satellite observation.

Response: The text was changed as follows:

[Lines 123–124] "The NWP fields are remapped to the resolution of satellite imagery by linear interpolation. We use the NWP products that are closest in time to the satellite observations."

4. L140: Are ec and âU ÌŃ s Ìğ ec are obtained are SDS data of 'cloud_emiss11_1km' and 'cloud_emission12_1km' as expressed in L200. Are they the emissivity or emissivity uncertainties?

Response: The ec values from C6 MYD06, of which SDS data are 'cloud_emiss11_1km' and 'cloud_emiss12_1km', are cloud emissivity itself, not cloud emissivity uncertainties. We clarify definitions of the data in the revised version.

[lines 253–257] "Here we use the cloud emissivity values at 11 and 12-$\mu$m for each ice cloud pixel provided in MYD06, for which the Scientific Data Set (SDS) names are 'cloud_emiss11_1km' and 'cloud_emiss12_1km'. The cloud emissivity for a single band is obtained by the following equation: $e\_c=(I\_obs-I\_clr)/(I\_ac+T\_ac\ B(T\_c)-I\_clr)$. (7) In Eq. (7), $T\_ac$ and $I\_ac$ are the above-cloud transmittance and the above-cloud emission (Baum et al., 2012), which are additional terms compared to the definition of the cloud emissivity in the infrared window regions in this paper (Eq. (2)). In spite of different definition of Eq. (7) from the Eq. (2), we use this cloud emissivity data since there the differences are small from the two different equations in the infrared window region."

5. L140:L155: Make this section clear and easy to understand. For example, how do you constrain 11 micron cloud emissivity for an ice cloud pixel (L147), and how do you use this information with LUT values?

Response: We constrain 11-$\mu$m cloud emissivity from minimum to maximum values for an ice cloud pixel that are provided in the LUT values. We modified a paragraph located at lines 140–155 in the original manuscript, as below.

[lines 185–201] "In this study, we apply a range of spectral cloud emissivity values to infer cloud temperatures rather than an optimum value. In our approach, the cloud is considered as a number of plane parallel homogeneous cloud layers. The cloud layer temperature ranges, Tc, are estimated as a vector of possible Tc values given a range of the ec and $\Delta$ec (hereafter, ec and $\Delta$ec) such as ec = [e_cˆ1,e_cˆ2,⋯,e_cˆn] and $\Delta$ec =[$\Delta$e_cˆ1,$\Delta$e_cˆ2,⋯,ãĂŰ$\Delta$eãĂŮ_cˆn ] as shown in Fig. 2(b). The ec and $\Delta$ec in Fig. 2(b) describes a range of possible spectral cloud emissivity values that can simulate the measured channel radiances. Thus, this study aims to produce Tc given the ec and $\Delta$ec, and to examine how closely the retrieved Tc are to the actual vertical cloud structure. The differences between this study and Inoue (1985) are summarized as follows. Constraints in the iteration range for cloud emissivity are provided in look-up tables (LUTs) discussed in the next section, as opposed to considering the full range of possible values from 0 to 1. Emissivity differences ($\Delta$ec) are used, rather than a single value for the extinction coefficient ratio between two infrared channels. Given the range of emissivity differences ($\Delta$ec provided in LUTs), we obtain a range of Tc (and hence a range of cloud heights, Hc) that can be compared to CALIPSO products. The first step in the current method (Fig. 3) is to constrain 11-$\mu$m cloud emissivity ranges (ec|11) that an ice cloud pixel can have based on the brightness temperatures. To obtain a reasonable ec|11 boundary corresponding to the ice cloud microphysical properties, the LUTs are generated to provide ec|11 ranges characterized by brightness temperature (BT) for 11 $\mu$m (BT|11), BT differences (or BTD) between 11 and 13 $\mu$m (BTD|11, 13) and between 11 and 12 (BTD|11, 12) (the light gray box in Fig. 3)."

6. L197-L204: This paragraph is also confusing. The first line of this paragraph states that you derive an empirical relationship, however, the last section discusses about taking per centile values. Do you use empirical relationship or percentile values to define the minimum and maximum values of the emissivity?

Response: We removed the expression, 'empirical relationship' to avoid giving confusions to readers. Also we revised this paragraph located at L197–204 in the original

manuscript.

[lines 253–267] "The final step is to find the possible ranges of ec and $\Delta$ec in each of the bins of BTD|11,13, BTD|11,12, and BT|11. Here we use the cloud emissivity values at 11 and 12-$\mu$m for each ice cloud pixel provided in MYD06, for which the Scientific Data Set (SDS) names are 'cloud_emiss11_1km' and 'cloud_emiss12_1km'. The cloud emissivity for a single band is obtained by the following equation: e_c=(I_obs-I_clr)/(I_ac+T_ac B(T_c )-I_clr). (7) In Eq. (7), T_ac and I_ac are the above-cloud transmittance and the above-cloud emission (Baum et al., 2012), which are additional terms compared to the definition of the cloud emissivity in the infrared window regions in this paper (Eq. (2)). In spite of different definition of Eq. (7) from the Eq. (2), we use this cloud emissivity data since there the differences are small from the two different equations in the infrared window region. Note that the cloud emissivity data from C6 MYD06 are retrieved under the assumption of the single-layered cloud. Here the possible ranges of ec and $\Delta$ec are determined as the min/max(ec) and ($\Delta$ec) among cloud emissivity values allocated by the bins of three parameters. To exclude extreme values, the min/max(ec) and ($\Delta$ec) are defined as the 2nd /98th percentiles of the ec and $\Delta$ec distributions when there are at least 5,000 pixels available for a given bin. When there are between 500 and 5000 pixels, the 5th /95th percentiles are chosen as the min/max(ec) and ($\Delta$ec). In the rare case when there are between only 200 and 500 pixels, the 10th /90th percentiles are used. Any case with fewer than 200 ice cloud pixels is not included in the LUTs."

7. Subsections 3.1 and 3.2 may be moved to data section.

Response: We moved subsections 3.1 and 3.2 to subsections 2.1 and 2.5 under the section 2 (the Data section).

8. L297: A brief description regarding the procedure of collocating CALIOP and MODIS is useful here.

Response: Done as follows:

[Figure]

[lines 353-354] "The computationally efficient method of Nagle et al. (2009) is used to collocate the simultaneous nadir observations (SNO) between two satellites. Following their approach, CALIOP is projected onto MODIS."

9. What are authors' view for deviated CBH and min(Hc)?

Response: The vector, Hc, provides a possible range of cloud heights for the observed channel radiances. When comparing to a lidar-based CBH, we also have to take note that the lidar signal attenuates as the COT increases.

10. Why not to write min_CTH or similar instead of min(Hc) ? Same for max (Hc) as well.

Response: Again, our products, Hc, provide a possible range of cloud heights for the observed channel radiances. Hc is not exactly same as the definition of cloud top height (CTH) or cloud base height(CBH).

11. The discussion of section 4 may be strengthen by referring past studies and/or putting authors' own logic.

Response: We added and modified the section 5 (section 4 in the original version of the manuscript). Most parts that were modified are paragraphs at lines 391–408, as below.

[Lines 404–421] "To better understand the potential biases of the current algorithm in comparison with CALIOP, we compare the mean(Hc) to the mean(CALIOP Hc) that are defined as $0.5 \cdot (\max(Hc)+\min(Hc))$ and as $0.5 \cdot (CALIOP\ CTH + CALIOP\ CBH)$, respectively. Fig. 10 shows the frequency of occurrence of biases, that is, the mean(CALIOP Hc) minus the mean(Hc), as a function of CALIOP COT for the single-layer ice clouds during August 2015. In a comparison of the MODIS cloud mask with CALIOP, Ackerman et al., (2008) noted that the cloud mask performs best at optical thicknesses above about 0.4. The lidar has a greater sensitivity to particles in a column than passive radiance measurements. Based on this consideration, we limited our results to

those pixels where the COT $\geq$ 0.5 in x-axis of Fig. 10. Fig.10 illustrates that our resulting single-layer ice clouds boundaries are consistent with CALIOP measurements, showing slightly negative biases except the region near 'COT$\leq$1.5'. These results suggest that our approach for applying a range of cloud emissivity values to estimate cloud boundaries has potential merit for using IR channels to produce cloud boundaries similar to those that the lidar observes, especially for optically thin but geometrically thick ice clouds which tend to have large uncertainties (Hamann et al., 2014). The negative biases of the mean(Hc) from CALIOP measurements are caused primarily by two factors: (1) The min(Hc) values for all cloud regimes tend to be higher than geometric cloud base, and (2) The max(Hc) values are sometimes slightly outside the actual cloud boundaries. Perhaps this is caused in part by the conversion of temperature to height using the NWP model product. Another source of error could be that the radiances have some amount of uncertainty that was not considered in our methodology. The notable point is that the boundary heights for optically thin cirrus (1.5<COT$\leq$3.5) show the lowest biases."

12. It is better to show the dependence of CTH or CBH difference between CALIOP and this study on CALIOP COT in Figure 10 instead of the mean value difference. What information do authors want to convey from the difference of C2 mean values?

Response: We understand your point here but we want to keep the figure the same for these reasons: (1) Fig. 10 would become more confusing if we show the dependence of the both of CTH and CBH difference between CALIOP. (2) What we intended to show is that for optically thin, but geometrically thick ice clouds, our cloud boundaries are consistent with CALIOP measurements. (3) As the results are shown for a limited optical thickness range, we feel it would not provide greater insight but would lead to a longer explanation necessary for understanding the min/max(Hc).

13. Figure 1:Make the caption clear. Write about Iclr and B in the caption.

Response: Done as below.

[20 pp.] "Figure 2: The conceptual model for (a) a plane parallel homogeneous cloud layer with no scattering, characterized by cloud emissivity (ec) and cloud emissivity differences between two infrared channels ($\Delta$ec) at the cloud temperature (Tc) and (b) a number of plane parallel homogeneous cloud layers (the stripes box) with a possible range of ec and $\Delta$ec such as ec = [e_cˆ1,e_cˆ2, ⋯,e_cˆn] and $\Delta$ec = [ãĂŰ$\Delta$eãĂŮ_cˆ1,ãĂŰ$\Delta$eãĂŮ_cˆ2, ⋯,$\Delta$e_cˆn] corresponding to a possible range of cloud temperature, Tc = [T_cˆ1,T_cˆ2, ⋯,T_cˆn], where Iclr and B are the clear-sky radiance and the Planck's function, respectively. Arrows represent upwelling radiances."

14. Figure 2: 'The logo of Copernicus Publications' should be removed from the caption. Response: We removed the 'The logo of Copernicus Publications' from the caption.

[21 pp.] "Figure 3: A flowchart for estimation of Tc and Hc corresponding to ec (from a light gray box that will be shown in Fig. 3) and $\Delta$ec (from a dark gray box that will be shown in Fig. 4) which represent cloud microphysics uncertainty in a certain cloud thickness. We denoted functions for minimum/maximum values of a matrix, A, as min/max(A)."   15. If COT is not used here, why do you use COT for y-axis title? Response: We removed the COT for y-axis title at the right side in Fig. 8, as below.

16. Table 1: What is IR cloud phase here?

Response: modified 'IR cloud phase' to the official name, 'IR cloud thermodynamic phase' (Baum et al., 2002), also added the reference for the product in the manuscript.

[lines 242–243] "Ice cloud pixels are identified by the MODIS IR cloud thermodynamic phase product in MYD06 (Baum et al. 2012) and where the pixels have a cloud top temperature $\leq$ 260K."

17. Table 2: Why 700 and 705 appear in this table?

Response: This is a systematic problem, a conversion from 'word' to 'PDF'. We corrected that problem in the revised manuscript.

Please also note the supplement to this comment:
https://www.atmos-meas-tech-discuss.net/amt-2019-148/amt-2019-148-AC4-
supplement.pdf